 **eLIFE**

# Motor selection dynamics in FEF explain the reaction time variance of saccades to single targets

Christopher K Hauser, Dantong Zhu, Terrence R Stanford, Emilio Salinas*

Department of Neurobiology and Anatomy, Wake Forest School of Medicine, Winston-Salem, United States

**Abstract** In studies of voluntary movement, a most elemental quantity is the reaction time (RT) between the onset of a visual stimulus and a saccade toward it. However, this RT demonstrates extremely high variability which, in spite of extensive research, remains unexplained. It is well established that, when a visual target appears, oculomotor activity gradually builds up until a critical level is reached, at which point a saccade is triggered. Here, based on computational work and single-neuron recordings from monkey frontal eye field (FEF), we show that this rise-to-threshold process starts from a dynamic initial state that already contains other incipient, internally driven motor plans, which compete with the target-driven activity to varying degrees. The ensuing conflict resolution process, which manifests in subtle covariations between baseline activity, build-up rate, and threshold, consists of fundamentally deterministic interactions, and explains the observed RT distributions while invoking only a small amount of intrinsic randomness.
DOI: https://doi.org/10.7554/eLife.33456.001

## Introduction

The reaction time (**RT**) represents the total time taken to perform all of the mental operations that may contribute to a particular action, such as stimulus detection, attention, working memory, or motor preparation. Although the importance of the RT as a fundamental metric for inferring the mechanisms that mediate cognition cannot be overstated (*Welford, 1980*; *Meyer et al., 1988*), such reliance is a double-edged sword. Under appropriate experimental conditions, differential measurements of RT may be used as a readout for changes in the (mean) time consumed by any one of the aforementioned operations, but a particular RT value is hard to interpret because it may be that not all of the operations involved are known, and those that are relevant may overlap in time to varying degrees. Furthermore, each operation may have its own, independent source of variability, making it very difficult to attribute the measured variance in RT to a particular cause (e.g., *Krajbich et al., 2015*). In the case of eye movements, this ambiguity is likely to be more severe than previously appreciated. There is a firm mechanistic account that describes how saccades are triggered, but according to the present results, that account lacks a crucial ingredient — ongoing motor conflict — and assumes, incorrectly, that in response to the same stimulus, the fundamental reason why some saccades are triggered very quickly whereas others take much longer simply boils down to noise in the underlying neuronal activity.

The neural dynamics that give rise to eye movements are well established. In essence, the preparation to make a saccade of a particular direction and amplitude is equal to a gradual rise in the activity of oculomotor neurons that are selective for the corresponding movement vector. If this rising activity, referred to as a motor plan, ramps up rapidly, the saccade is initiated quickly; if the motor plan grows more slowly, the saccade starts later. Quantitatively, this corresponds to a negative correlation between saccadic RT and build-up rate (*Hanes and Schall, 1996*; *Fecteau and*

*For correspondence:
esalinas@wakehealth.edu

**eLife digest** As we examine the space around us our eyes move in short steps, looking toward a new location about four times a second. Neurons in a region of the brain called the frontal eye field help initiate these eye movements, which are known as saccades. Each neuron contributes to a saccade with a specific direction and size. Before a saccade, the relevant neurons in the frontal eye field steadily increase their activity. When this activity reaches a critical threshold, the visual system issues a command to move the eyes in the appropriate direction. So a saccade that moves the eyes to the right requires a specific group of neurons to be strongly activated – but, at the same time, the neurons responsible for movement to the left need to be less active.

Imagine that you have to move your eyes as quickly as possible to look at a spot of light that appears on a screen. Some of the time your eyes will start to move about 100 milliseconds after the light appears. But on other attempts, your eyes will not start moving until 300 milliseconds after the light came on. What causes this variability?

To find out, Hauser et al. recorded from neurons in monkeys trained to perform such a task. When the spot of light appeared many different neurons were active, suggesting there is conflict between the plan that would move the eyes toward the target and plans to look at other locations. That is, when the target appears, the monkey is already thinking of looking somewhere. The time required to resolve this conflict depends on how far apart the target and the competing locations are from one another, and on how much the competing neurons have increased their activity before the target appears.

Similar mechanisms are likely to operate when we sit at the dinner table and look for the salt shaker, for example, and so the results presented by Hauser et al. will help us to understand how we direct our attention to different points in space. Understanding how these processes work in more detail will help us to discern what happens when they go wrong, as occurs in attention deficit disorders like ADHD.

DOI: https://doi.org/10.7554/eLife.33456.002

*Munoz, 2007*; *Ding and Gold, 2012*; *Heitz and Schall, 2012*; *Costello et al., 2013*). Notably, neurons that encode motor plans seem to reach a consistent firing level just before the onset of a saccade, particularly in the superior colliculus (**SC**) and the frontal eye field (**FEF**) (*Hanes and Schall, 1996*; *Brown et al., 2008*; *Stanford et al., 2010*; *Ding and Gold, 2012*). That is, there appears to be a fixed activity threshold that serves as a trigger for eye movements (*Lo and Wang, 2006*). Thus, it is widely thought that, for simple saccades to lone, unambiguous stimuli, i.e., reactive saccades, the variance of the RT distribution is predominantly determined by the variance of the FEF/SC build-up rates across trials (*Carpenter and Williams, 1995*; *Hanes and Schall, 1996*; *Fecteau and Munoz, 2007*; *Sumner, 2011*).

This rise-to-threshold process is of enormous conceptual importance, as it is the key building block of virtually all models of decision-making in which multiple-choice alternatives, typically guided by perceptual information, are evaluated over time (*Gold and Shadlen, 2001*; *Erlhagen and Schöner, 2002*; *Smith and Ratcliff, 2004*; *Brown and Heathcote, 2008*; *Stanford et al., 2010*; *Krajbich and Rangel, 2011*; *Thura et al., 2012*; *Brunton et al., 2013*; *Miller and Katz, 2013*). Nevertheless, the variance of this process in its simplest possible instantiation — reactive saccades — remains a mystery (*Sumner, 2011*), because it seems too large to reflect noise or intrinsic randomness (in the build-up rates of oculomotor neurons) alone. One possibility is that the randomness is purposeful, that unpredictability in saccade timing somehow entails a behavioral advantage (*Carpenter, 1999*). Alternatively, the RT of reactive saccades may fluctuate, at least in part, because of underlying neural mechanisms that have simply not been identified yet.

Here we describe such mechanisms. We recorded activity from single FEF neurons in an elegant paradigm (*Lauwereyns et al., 2002*; *Hikosaka et al., 2006*) that produces a large spread in saccadic RT simply by varying the subject's spatial expectation of reward. A model based on competitive dynamics quantitatively reproduced the temporal profiles of the evoked neural responses, as well as their dependencies on RT, reward expectation, and trial outcome (correct/incorrect) — this, while simultaneously matching the monkeys' full RT distributions across experimental conditions. The

results indicate that RTs vary because the stimulus-driven activity does not start from a consistent, neutral state, but rather from a dynamic, biased state in which incipient, internally driven motor plans are already developing. In other words, when the target appears, the monkey is already thinking of looking somewhere. This conflict between motor alternatives (1) requires varying amounts of time to be resolved, depending on how advanced and how congruent the bias-driven plans are relative to the target-driven response, (2) is initially defined by the baseline levels of activity (measured during fixation) across spatial locations, and (3) dictates the build-up rate and threshold of the subsequent rise-to-threshold process. Thus, the noise in the build-up rate is much more modest than that predicted by extant frameworks, and the high variability of saccades to single targets is, to a large degree, deterministic, a direct consequence of motor selection mechanisms that allow voluntary saccades to be driven by both sensory events and internal biases.

## Results

### Behavioral manifestations of a spatial bias

Two rhesus monkeys were trained on the one-direction rewarded (**1DR**) task (*Figure 1a*), in which a saccade to a lone, unambiguous target must be made but a large liquid reward (primary reinforcer) is available only when the target appears in one specific location (*Lauwereyns et al., 2002*; *Hikosaka et al., 2006*). The rewarded location remains constant over a block of trials and then changes. Of note, ours is a RT version of the task whereby the go signal (offset of fixation point), which means 'move now!' is simultaneous with target onset. Also, it involves four locations and variable block length. This task generates errors and a large spread in RT (*Figure 2*) under minimalistic sensory stimulation conditions. We exploit this to investigate how variance in saccadic performance relates to variance in FEF activity.

When the target and rewarded locations coincided (congruent trials; *Figure 2a*, red traces), the monkeys consistently moved their eyes very quickly (monkey G, $158 \pm 33$ ms, mean RT $\pm$ 1 SD; monkey K, $146 \pm 21$ ms), and essentially never missed (monkey G, 99.8% correct, $n = 7234$ congruent

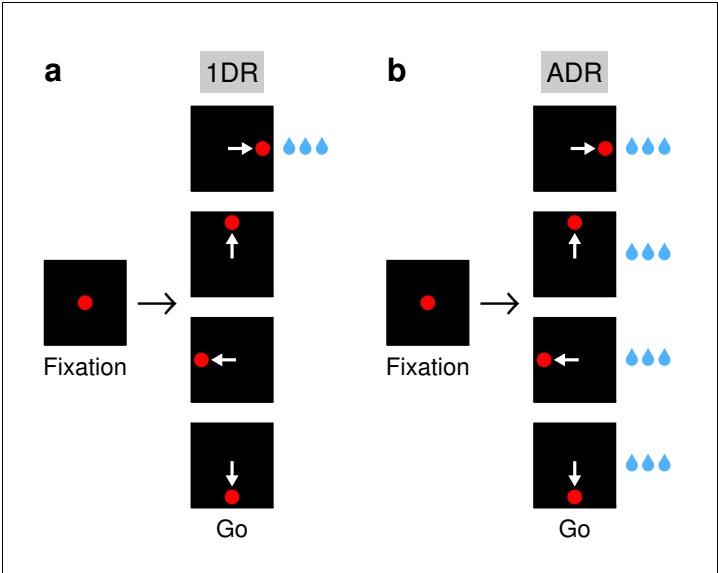

**Figure 1.** Saccadic tasks used. (**a**) The 1DR task. After a fixation period of 1000 ms, a single eccentric stimulus appears at one of four locations and the subject is required to make a saccade to it. Stimulus location is chosen randomly in each trial. Fixation offset (go signal) and stimulus onset are simultaneous. In each block of trials, only one of the directions yields a large reward; the others yield either no reward (monkey G) or a small reward (monkey K). (**b**) The ADR task. Same sequence of events as in (**a**), except that saccades in all directions are equally rewarded. White arrows indicate saccades; they are not displayed.
DOI: https://doi.org/10.7554/eLife.33456.003

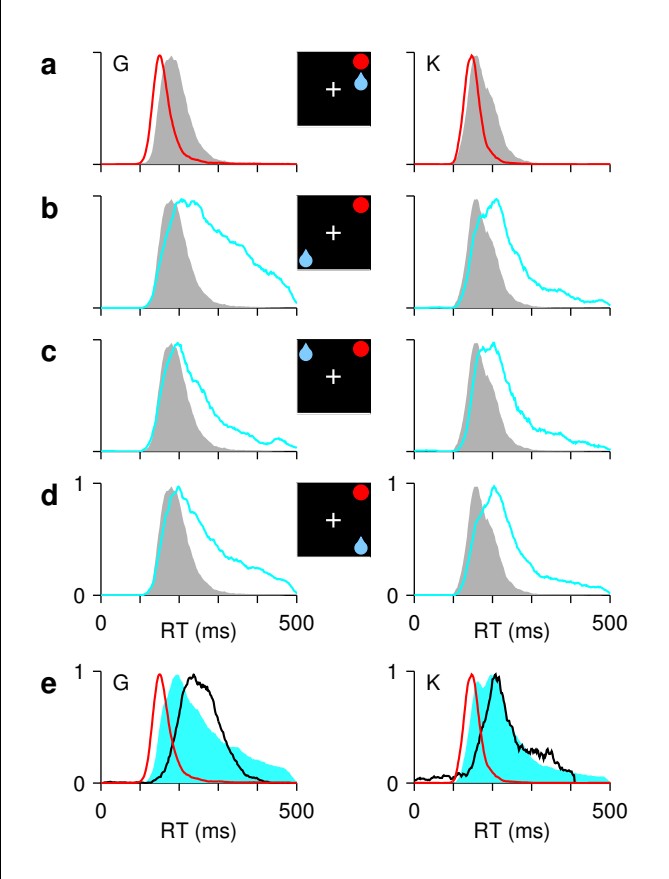

**Figure 2.** Asymmetric reward expectation leads to strong spatial bias. (**a–d**) RT distributions for correct saccades, for monkeys G (left column) and K (right column). Insets indicate rewarded location (blue drop) and target (red circle). When the two are congruent ([**a**], red traces), RTs are shorter and less variable than when they are incongruent, that is, either opposite ([**b**], cyan) or adjacent ([**c**], [**d**], cyan). In unbiased trials (ADR task; gray), results are intermediate. (**e**) RT distributions in correct congruent (red, same data as in [**a**]), correct incongruent (cyan, data in [**b-d**] combined), and incorrect incongruent (black) trials. Histograms are normalized to a maximum of 1. The RTs during errors are neither the fastest nor the slowest.

DOI: https://doi.org/10.7554/eLife.33456.004
The following figure supplements are available for figure 2:

**Figure supplement 1.** Variations in spatial bias over time.
DOI: https://doi.org/10.7554/eLife.33456.005
**Figure supplement 2.** Impact of reward-location bias on saccade metrics.
DOI: https://doi.org/10.7554/eLife.33456.006

trials; monkey K, 99.6%, $n = 5837$). By contrast, when the rewarded and target locations were either diametrically opposed or adjacent (incongruent trials; *Figure 2b–d*, cyan traces), both the mean RT and the spread increased dramatically (monkey G, $269 \pm 84$ ms; monkey K, $236 \pm 77$ ms), as did the percentage of incorrect saccades away from the target (monkey G, 18.3% incorrect, $n = 16905$ incongruent trials; monkey K, 8.1%, $n = 12708$). The symmetric condition in which all directions were equally rewarded (**ADR**; *Figure 1b*) produced RT distributions that were intermediate between those of congruent and incongruent trials (*Figure 2a–d*, gray; monkey G, $192 \pm 40$ ms; monkey K, $174 \pm 36$ ms). These results recapitulate the puzzle mentioned in the Introduction: if saccades can be very fast, why, under identical stimulation conditions, are they sometimes very slow?

Also note that, compared to those of correct saccades, the RTs of incorrect saccades were neither consistently fast, as might be expected on the basis of strong anticipation, nor consistently slow, as might be expected from a protracted deliberation process (*Figure 2e*). Instead, the RTs during errors fell squarely in the middle of the distributions of correct RTs (for correct trials, 90% of RTs

were inside the ranges 158–432 ms and 146–404 ms for monkeys G and K, respectively; for errors, the ranges were 180–336 and 152–388 ms). Mechanistically, it is not obvious how this could be accomplished. One of the aims of the model presented below is to reproduce the RT data shown in *Figure 2* and to resolve this conundrum.

In short, monkeys are highly sensitive to the spatially asymmetric value associated with otherwise identical target stimuli (for additional evidence of this, see *Figure 2—figure supplement 1*). This manifests primarily as large differences in RT between correct congruent, correct incongruent, and incorrect incongruent trials (for other manifestations, see *Figure 2—figure supplement 2*). The pattern of results suggests that, when a target appears at an unrewarded location, a conflict arises because the monkey wants to look at the rewarded location instead. In what follows, we ask: what features of the evoked FEF activity reflect such conflict, and do they account for the measured variability in the direction and timing of the elicited saccades?

## Motor conflict during the rise-to-threshold process

We recorded single-unit activity from 132 FEF neurons in the two monkeys (67 in monkey G; 65 in monkey K) during performance of the 1DR task. The analysis in this section focuses on a population of 62 neurons that satisfied two conditions: they had standard visual and saccade-related responses, and both correct and incorrect trials were collected for them (for details of cell and trial selection procedures, see 'Materials and methods, *Neuronal classification and selection*').

In this section, we show that, during the rise-to-threshold process associated with reactive saccades, the build-up rate, the apparent threshold, and the baseline firing rates measured during fixation (before the target is presented) exhibit coordinated variations that have not been appreciated before and which, beyond the specifics of our task, are likely to be key for determining RTs in general. We refer to the baseline levels in plural because the target location is not the only relevant one; neurons with RFs at other locations may be activated by internally driven biases, creating motor conflict, that is, activity that competes with and impacts the target-driven response — and the RT.

Before describing these results, a note about nomenclature. Whereas just two distinct experimental conditions, congruent and incongruent, are relevant for behavioral analysis, neurophysiologically there are eight conditions to consider, depending on whether the target, expected reward, and saccadic movement were inside or outside a recorded cell's response field (**RF**). Thus, for example, we use the abbreviation IOI to denote the target/reward/saccade combination in/out/in; that is, trials for which the target was in, the reward was expected out, and the saccade was into the RF (see icons in *Figure 3a–c*). Such IOI trials are incongruent, because target and reward locations do not match, and correct, because the target and saccade locations match. Only six of the eight possible combinations are considered because congruent trials were virtually devoid of errors, so IIO and OOI conditions are absent. With this notation at hand, we now turn to the activity of FEF across the six remaining conditions.

Although each saccade in the 1DR task involves a single target, the motor preparation process in FEF can (and should) be understood as a motor selection process involving not one but at least two populations of neurons, those that contribute to the actual saccadic choice and those that favor the opposite choice (*Figure 3a–c*, see icons). During congruent trials, when the target appears at the rewarded location, the evoked activity rises most rapidly and reaches the highest firing rate (*Figure 3a*, III trials, red trace). The saccade is essentially always correct and no evidence of conflict is discernible because the neurons favoring the opposite choice, away from the target, show little (if any) response before the eye movement (*Figure 3a*, OOO trials, green trace). In this case, naively, it would appear as if only the one response that rises to threshold is important.

Notably, though, a difference in activity between the two complementary populations is already evident before the go signal/target onset. This is the internal bias signal created by reward expectation, which in this case is spatially congruent with both the target and the saccade. This baseline signal is consistent with previous neurophysiological studies using the 1DR task (*Takikawa et al., 2002*; *Sato and Hikosaka, 2002*; *Ding and Hikosaka, 2006*), and may be interpreted as a neural correlate of spatial attention (*Maunsell, 2004*; *Peck et al., 2009*; *Preciado et al., 2017*).

By contrast, during incongruent trials, when the target is presented opposite to the rewarded location, a conflict arises early in the trial in the form of a higher baseline favoring the rewarded location (*Figure 3b*, note green trace above red before go signal). During correct trials this conflict is appropriately resolved as the target-driven activity increases and overtakes the competition

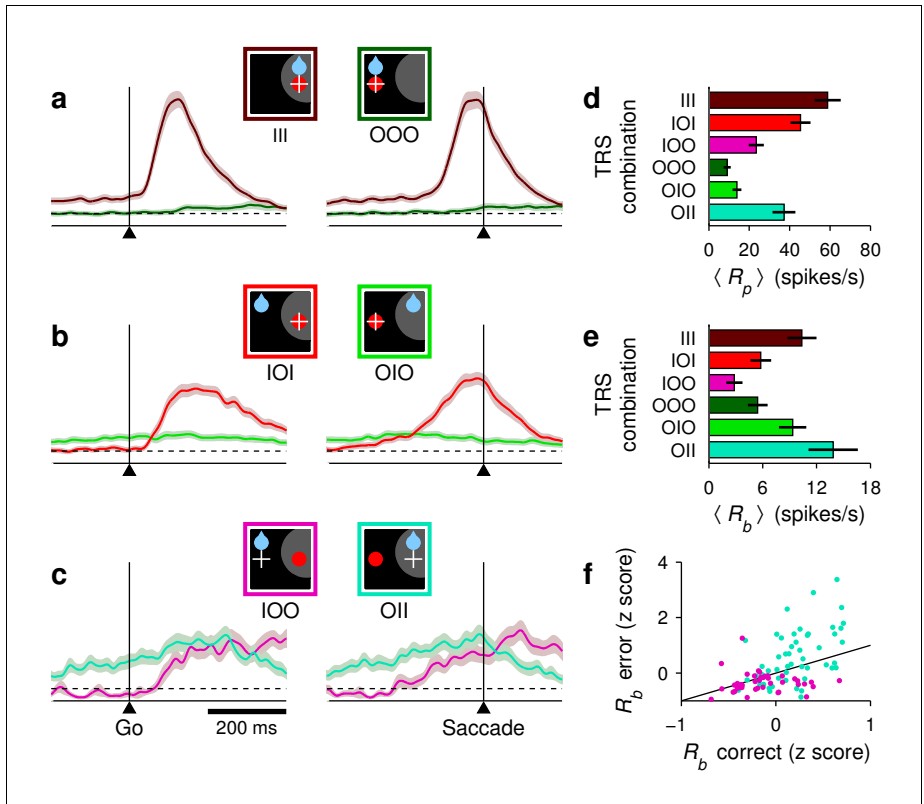

**Figure 3.** Baseline activity predicts response gain, threshold, and outcome. (a–c) Normalized firing rate as a function of time for a population of 62 neurons (V, VM, and M) for which both correct and incorrect responses were collected. Icons indicate target (red circle), rewarded location (blue drop), and saccade (white cross) relative to the RF (gray area) in each case. Paired reddish and greenish traces correspond to activity with the target inside or outside the RF, respectively, in the same behavioral condition. For congruent trials (a) only correct responses are shown. For incongruent trials both correct (b) and incorrect (c) responses are shown. The reference line (dotted) is identical across panels. (d) (e) Mean peak activity (d) and mean baseline activity (e) for each target-reward-saccade (TRS) combination, from the same 62 neurons in (a–c). Error bars indicate ±1 SE across cells. (f) Baseline activity (z-scored) for incorrect (y axis) versus correct outcomes (x axis) under identical target and reward conditions. Each point is one neuron. Magenta dots indicate IOO versus IOI trials (target in/reward out); cyan dots indicate OII versus OIO trials (target out/reward in).

DOI: https://doi.org/10.7554/eLife.33456.007

The following figure supplement is available for figure 3:

**Figure supplement 1.** Responses of predominantly visual and predominantly movement-related neurons in FEF across conditions and outcomes.

DOI: https://doi.org/10.7554/eLife.33456.008

(*Figure 3b*, IOI trials, red trace), but the rise proceeds more slowly, that is, it has a lower build-up rate and ultimately reaches a lower peak firing level ($R_p$; see 'Materials and methods, *Peak response*') than that observed when the bias and the target are congruent (*Figure 3d*, IOI vs. III). Finally, the conflict is even more extreme during incongruent trials that end in erroneous choices toward the rewarded location (*Figure 3c*). In that case, the initial bias in baseline activity is maximized (*Figure 3e*, IOO vs. OII) and the evoked target-driven activity (*Figure 3c*, magenta trace) is considerably weaker than that observed during correct saccades (*Figure 3d*, IOO vs. IOI). The neural response associated with the (wrong) saccadic choice rises very slowly (*Figure 3c*, cyan trace) and reaches a modest threshold level prior to saccade onset, but this activity is nonetheless slightly above that associated with the opposite (correct) motor alternative (*Figure 3c*, right panel; *Figure 3d*, OII vs. IOO). This ambivalence between motor plans is reminiscent of that observed during choice tasks (*Thompson et al., 2005*; *Ding and Gold, 2012*; *Costello et al., 2013*), as if the

monkeys had struggled to make a choice between two competing targets, even though only one was displayed.

These results were based on recordings from 62 neurons with diverse visuomotor properties, but were qualitatively similar when the averaging across cells was restricted to units that were either predominantly visual or predominantly motor (*Figure 3—figure supplement 1*). Those two populations differed in the time at which they fired maximally and in the magnitude of their baseline activity, but qualitatively, their responses changed similarly across conditions and outcomes.

The mean peak and baseline responses, $R_p$ and $R_b$, demonstrated remarkably similar variation patterns (*Figure 3*, compare [d] and [e]). Overall, these comparisons based on population responses suggest that the baseline firing rates (at both the target and opposite spatial locations), build-up rate, and threshold of the rise-to-threshold process vary in a coordinated way across experimental conditions. But then a crucial question arises: do such covariations also occur from trial to trial *within* each condition? We sought evidence of this by analyzing the responses of individual neurons.

First we found that, during incongruent trials, the baseline activity, $R_b$ (firing rate in a 250 ms window preceding target onset), is strongly predictive of outcome. When the rewarded location coincided with the RF and the target was presented outside, most neurons (34 of 53, $p = 0.03$, binomial test) had a higher baseline rate before incorrect as opposed to correct trials (*Figure 3f*, cyan dots, OII vs. OIO; $p = 0.002$, permutation test), as if an excessive $R_b$ triggered a (wrong) saccade into the RF. Conversely, when the rewarded location was opposite to the RF and the target was subsequently presented inside, most neurons (33 of 42, $p = 0.001$, binomial test) had a lower $R_b$ preceding incorrect trials (*Figure 3f*, magenta dots, IOO vs. IOI; $p = 0.009$), as if the lack of baseline activity precluded a (correct) saccade into the RF.

Second, for each recorded cell (V, VM, and M), we calculated Spearman correlation coefficients between pairs of neural response measures across trials ('Materials and methods, *Statistical analyses*'). Although some of those correlations were too noisy to resolve (e.g., between baseline activity and build-up rate), that between baseline activity and peak response, $\rho(R_b, R_p)$, was highly robust; it was strongly positive not only in IOI trials (*Figure 4a*) but also in other conditions (III, OOO, and OIO; $p<0.001$ in all cases), and there was no evidence of a dependence on the visuomotor properties of the recorded cells (*Figure 4a*, scatterplot; $p>0.27$ for linear regression). This confirms that when

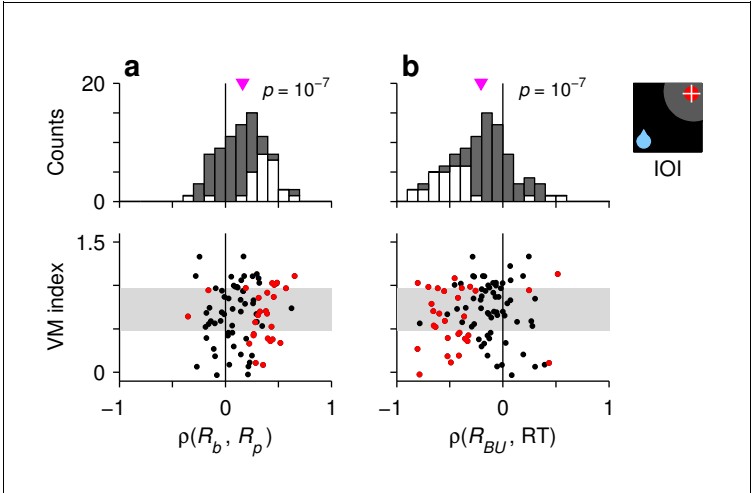

**Figure 4.** Characteristic manifestations of the rise-to-threshold process in single FEF neurons. Each panel shows Spearman correlation coefficients calculated for 90 V, VM, and M neurons. (**a**) Correlations between baseline activity and peak response. (**b**) Correlations between build up-rate and RTals. In each panel, the scatterplot (bottom) shows the same correlation coefficients as the histogram (top) but together with the visuomotor indices of the neurons. Gray shades in the scatterplots demarcate the borders defining the V, VM, and M cell categories. White bars in histograms and red points in scatterplots indicate significant correlations ($p<0.05$). Pink triangles mark mean values, with significance from a signed-rank test indicated.
DOI: https://doi.org/10.7554/eLife.33456.009

the baseline activity of a neuron is high, the response evoked later, after the target appears, typically reaches a higher firing level than when the baseline is low.

At the single-cell level, we also found strong associations between responsivity and RT. In IOI trials, the correlation between build-up rate and RT, $\rho(R_{BU}, \mathrm{RT})$, was strongly negative (*Figure 4b*). This is in agreement with the common finding reported by previous studies, that the build-up rate is the main link between oculomotor activity and saccade latency. Furthermore, the correlations $\rho(R_b, \mathrm{RT})$ and $\rho(R_p, \mathrm{RT})$ were also predominantly negative, consistent with the observation that, across conditions with movements into the RF, shorter RTs are simultaneously associated with higher baseline, higher build-up rate, and higher threshold (*Figure 3a,b*). These dependencies, and important deviations observed across trial types, are analyzed in detail below.

In summary, during the 1DR task, the activity in FEF demonstrated characteristic covariations in the three main features of the rising saccade-related activity: the 'baseline' firing rates at both the target and opposite spatial locations, the build-up rate of the target-driven response, and the maximum activity reached before movement onset. This suggests a causal relationship between the baseline and the subsequent response, because the baseline signal arises earlier (before target presentation) and because it is predictive of outcome (*Figure 3f*). Although these covariations were more evident across task conditions than across trials, our contention is that they are always present because they reflect fundamental dynamics of the oculomotor circuitry. Next, to test this, we present a mechanistic model that generalizes these novel interrelationships to all trials, and is thereby able to relate neuronal activity to RT with remarkable detail.

## Modeling the rise-to-threshold dynamics

A saccadic competition model was developed to investigate the mechanistic link between the FEF activity and the monkeys' behavior in the 1DR task ('Materials and methods, *Saccadic competition model*'). Such a bridge requires that multiple constraints are satisfied. First of all, the model must reproduce the neurophysiological results presented in the previous section. Thus, it considers two neural populations whose responses may rise to a threshold. One population generates saccades toward location $T$, where the target stimulus is presented, and the other toward $D$, the diametrically opposite location (*Figure 5a*, icon). In any given trial, if the target-driven response, $R_T$ (*Figure 5a–c*, red traces), reaches threshold first, a correct saccade to the target is produced, whereas if the bias-driven activity, $R_D$ (*Figure 5a–c*, blue traces), reaches threshold first, the result is an incorrect saccade away from the target. In this way, the $R_T$ and $R_D$ variables correspond to the population responses recorded with the target inside (*Figure 3a–c*, reddish traces) and outside (greenish traces) of the RF, respectively.

Another important feature of the recorded data is the evident asymmetry between the two motor plans: the target-driven activity is typically strong and never fully suppressed, whereas the internally driven activity favoring the opposite location is typically suppressed and only rarely of moderate strength. In the model, this asymmetry is captured by two suppression mechanisms that constrain when and how $R_D$ can rise. One of them ('Materials and methods, *Saccadic competition model*, Rule 1') simply prescribes that, once $R_T$ is rising, it can fully suppress $R_D$. That is, the moment $R_T$ advances past $R_D$, $R_D$ stops rising altogether. The other mechanism is about the timing of $R_D$. The target-driven response, $R_T$, begins to rise shortly after target onset (after an afferent delay of 35 ms), whereas its counterpart, $R_D$, begins to rise later, partly because of a somewhat longer afferent delay (50 ms) but mostly because of a transient, stimulus-driven suppression. This (partial) suppression is based on abundant evidence (reviewed by *Salinas and Stanford, (2018)*; see also *Reingold and Stampe (2002)*; *Dorris et al. (2007)*; *Stanford et al. (2010)*; *Bompas and Sumner (2011)*; *Buonocore and McIntosh (2012)*; *Buonocore et al., (2017)*) indicating that ongoing saccade plans, $R_D$ in this case, are briefly inhibited by stimuli that appear abruptly, just like the target in our experiment. In the model, the inhibition lasts 115 ms (*Figure 5a–c*, gray shades), after which $R_D$ may rise in full force — if it was not overtaken by $R_T$ in the interim. These two suppression mechanisms give the target-driven activity an advantage over its competing, internally driven counterpart, and they are necessary to reconcile the occurrence of errors with the late-onset, weak activity away from the RF seen during correct saccades (*Figure 3a,b*, greenish traces). For instance, without the transient, stimulus-driven inhibition, the model would produce incorrect saccades at a rate that is vastly higher than that observed experimentally.

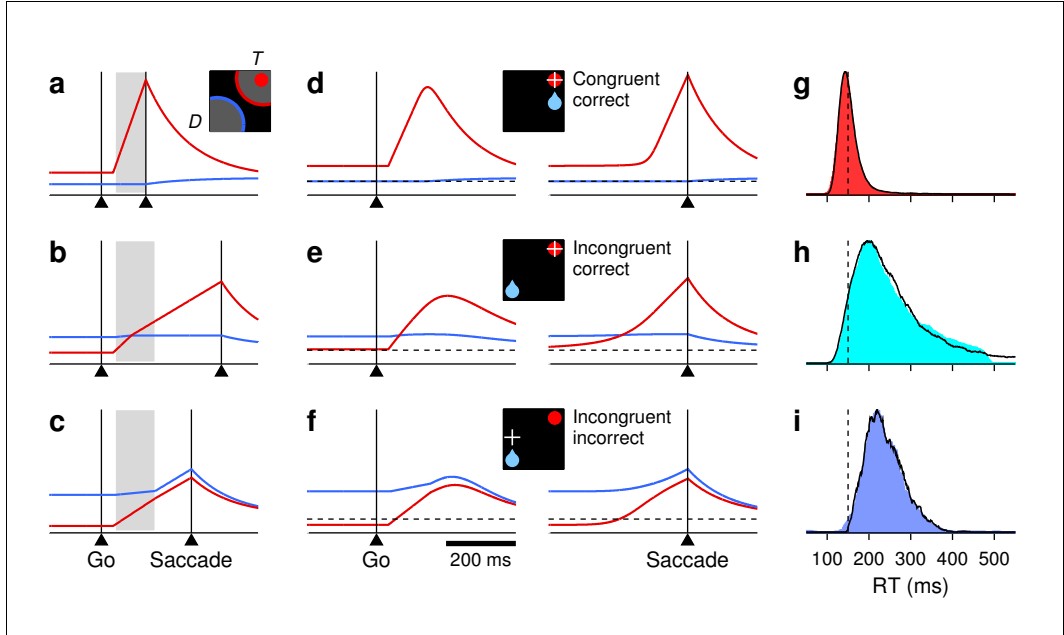

**Figure 5.** Simulation results from a saccadic competition model that bridges the neural and behavioral data. (a–c) Simulated activity in three single trials. Traces show motor plans toward the target location ($T$, red) and the diametrically opposite location ($D$, blue). Gray shades mark the period during which the target stimulus suppresses the motor plan toward $D$. When the target-driven activity, $R_T$ (red), reaches threshold first (a, b), the saccade is correct; when the bias-driven activity, $R_D$ (blue), reaches threshold first (c), the saccade is incorrect. The scale on the y axes is the same for all three plots. (d–f) Mean firing rate traces $R_T$ (red) and $R_D$ (blue) averaged across correct congruent (d), correct incongruent (e), and incorrect incongruent (f) simulated trials. Same format as in *Figure 3a–c*. (g–i) RT distributions from the same simulations as in panels (d–f), with trials sorted accordingly: for correct congruent (g), correct incongruent (h), and incorrect incongruent (i) trials. Colored shades are combined data from the two monkeys; black lines are model results. Dotted vertical lines at 150 ms are for reference.

DOI: https://doi.org/10.7554/eLife.33456.010

The following figure supplement is available for figure 5:

**Figure supplement 1.** Variations in baseline and threshold levels in FEF and in the saccadic competition model.
DOI: https://doi.org/10.7554/eLife.33456.011

Finally, the model must also capture the variations in baseline activity, build-up rate, and threshold observed in the FEF data (*Figures 3d,e* and *4a*), and this is where the crucial conceptual leap takes place. What we found empirically was that, for the target-driven response, a higher baseline was accompanied by both a higher build-up rate and a higher threshold, with the baseline activity of the alternative motor plan having opposite effects. The model generalizes these dependencies by making two assumptions. First, that similar relationships hold across *all trials*, rather than just across the three experimental conditions examined, and second, that because the baseline signal is present before target onset, any variations in build-up rate and threshold can be formulated mainly as the result of variations in baseline activity. Thus, the model can be thought of as designed to test whether the differences in the rise-to-threshold process observed across experimental conditions (*Figure 3*) are the average manifestations of similar but more general dynamical relationships between the three relevant variables, where the variance is primarily derived from the baselines. So, in practice, the general idea is that the baselines fluctuate stochastically and determine the ensuing rise-to-threshold excursion in each trial.

The resulting dynamics between competing motor plans can be intuitively appreciated with three example trials (*Figure 5a–c*). The simplest situation is when, during fixation, the baseline at the target location, $B_T$, is larger than that at the opposite location, $B_D$ (*Figure 5a*). This is typically the case when the target and rewarded locations coincide, but what matters in the model is simply the actual baseline values (more on this below). The condition $B_T > B_D$ has two specific consequences: (1) it yields a high build-up rate for the target-driven activity, $R_T$ (red trace), and (2) it sets the saccade

threshold, $\Theta$, to a high value (*Equations 6, 7*). Thus, because of the high build-up rate, $R_T$ rises sharply and quickly triggers a saccade, in spite of the high $\Theta$. The $D$ plan (blue trace) is always suppressed in this case, so no overt conflict is visible. This is how correct saccades with very short RTs are produced.

The more interesting scenario occurs when the bias-driven plan starts with the higher baseline, as typically happens when the reward is expected at the $D$ location (but again, the dynamics are dictated just by the baseline values). In that case, the saccade can be either correct or incorrect, depending on how big the lead is. When $B_D$ is much larger than $B_T$ (*Figure 5c*), the consequences are essentially the opposite of those in the previous example: (1) $R_T$ has a low build-up rate, so the target-driven response (red trace) rises slowly, and (2) the saccade threshold, $\Theta$, is low. In this way, $R_D$ is able to advance steadily after the suppression interval and win the race from wire to wire, reaching a relatively low firing level before saccade onset. This is how incorrect saccades are produced.

By contrast, if the baseline $B_D$ is only moderately higher than $B_T$ (*Figure 5b*), then the combined effect of the baselines is intermediate relative to that in the two previous examples: (1) the initial build-up rate of $R_T$ is moderate, neither as high as that in the first example nor as low as that in the second, and (2) the value of $\Theta$ is also intermediate. The target-driven plan (red trace) rises at a rate that allows it to overtake the competing plan (blue trace) and to win the race by coming from behind. Importantly, in this case, $R_T$ slows down as it goes past $R_D$ (note the slight change in the slope of the red trace during the shaded interval). Although $R_T$ wins the race, overtaking the competing plan exacts a toll, and the lower its initial build-up rate, the higher that toll (*Equation 11*). This is the one mechanism that was introduced into the model specifically to satisfy key behavioral constraints. In this case, slowing down the *winner* target-driven plan is necessary to produce correct saccades with very long RTs — longer than those of incorrect saccades.

These examples illustrate how the baseline levels $B_T$ and $B_D$ quantitatively regulate both the build-up rate of the target-driven activity and the saccade threshold. Nevertheless, it is important to stress that, at the same time, the baselines convey information about the location of the expected reward in a manner that is consistent with the experimental data. In the simulations, the baseline values across trials are characterized by their mean and variance, which are fixed (*Equation 5*). However, the two mean values are assigned to the $T$ and $D$ locations according to a simple prescription: the rewarded location gets the higher mean ('Materials and methods, *Saccadic competition model*'). Thus, in simulations of the congruent case, $B_T$ is typically — although not always — larger than $B_D$ (as in *Figure 5a*), and the majority of trials are fast and correct; whereas in simulations of the incongruent case, the roles are reversed, $B_T$ is (on average) lower than $B_D$ (as in *Figure 5b,c*), which results in a combination of correct and incorrect slower responses. Other than that, the simulations proceed in exactly the same way in the two bias conditions, as they should.

With all of these elements in place, the model parameters were adjusted to fit all of the experimental data discussed so far ('Materials and methods, *Correspondence between data and model parameters*'). In this way, when trials were sorted by bias and outcome, the simulated neural responses reproduced the covariations in baseline, build-up rate, and threshold across conditions (*Figure 5d–f*; for quantification, see *Figure 5—figure supplement 1*). This demonstrates that, as intended, the hypothesized coupling across trials is indeed consistent with the observed coupling across experimental conditions. In addition, the average $R_T(t)$ and $R_D(t)$ traces matched the trajectories of the recorded population responses in great detail (compare to *Figure 3a–c*). The proposed interaction mechanisms between the two motor plans resulted in average traces with the appropriate magnitude, time course, and degree of ambivalence. But most critically, at the same time, the model fully accounted for the behavioral data: (1) it generated correct and incorrect saccades in proportions similar to those found experimentally ($\sim 0\%$ and $\sim 10\%$ errors in congruent and incongruent conditions), and (2) it generated simulated distributions of RTs (*Figure 5g–i*) that closely mimic their behavioral counterparts (as assessed by mean, median, SD, and skewness). Note, in particular, that the RTs in incorrect trials (panel i) are neither too fast, because the stimulus-driven suppression mechanism prevents fast errors, nor too slow, because the slowest responses (which occur when $R_T$ slows down) are correct. The model explains the behavioral data in terms of the neural data, accurately replicating both.

The results show that, as an intrinsic part of the motor competition process, the baseline activity, build-up rate, and threshold vary in a coordinated fashion to generate the wide range of RTs

observed in the task. In the rest of the paper, we show that this fact explains many other, seemingly odd features of the neural data.

## Comparison accross conditions with equalized RTs

A key assumption of the model is that fluctuations in baseline activity result in fluctuations in build-up rate and threshold. To further characterize the interdependencies between these three variables and better understand how they impact the RT, we compared the responses evoked in congruent versus incongruent trials before and after equalizing their RTs.

The FEF responses recorded during III and IOI trials were quite distinct (*Figure 3a,b*), even though both involved correct saccades in the same direction. The differences could be due to the different expected reward locations in the two conditions and/or to the different RTs generated (*Figure 6a*). To eliminate the differences due to RT, we devised a simple sub-sampling procedure ('Materials and methods, *RT matching*') that resulted in IOI and III data subsets with identical numbers of trials and nearly identical RT distributions (*Figure 6b*). Then, we made comparisons across conditions with and without matching the RTs.

What should be expected on the basis of the model? In the standard case, without RT matching (NM condition), the target-driven response, $R_T$, has a higher baseline and reaches a higher threshold in congruent trials than in incongruent trials (*Figure 6c*). However, because they have very different build-up rates (note the steeper rise of the magenta curve), when aligned on saccade onset, the corresponding response trajectories intersect each other twice. By contrast, when their respective RT distributions are the same (YM condition), the shapes of the trajectories are much more similar and

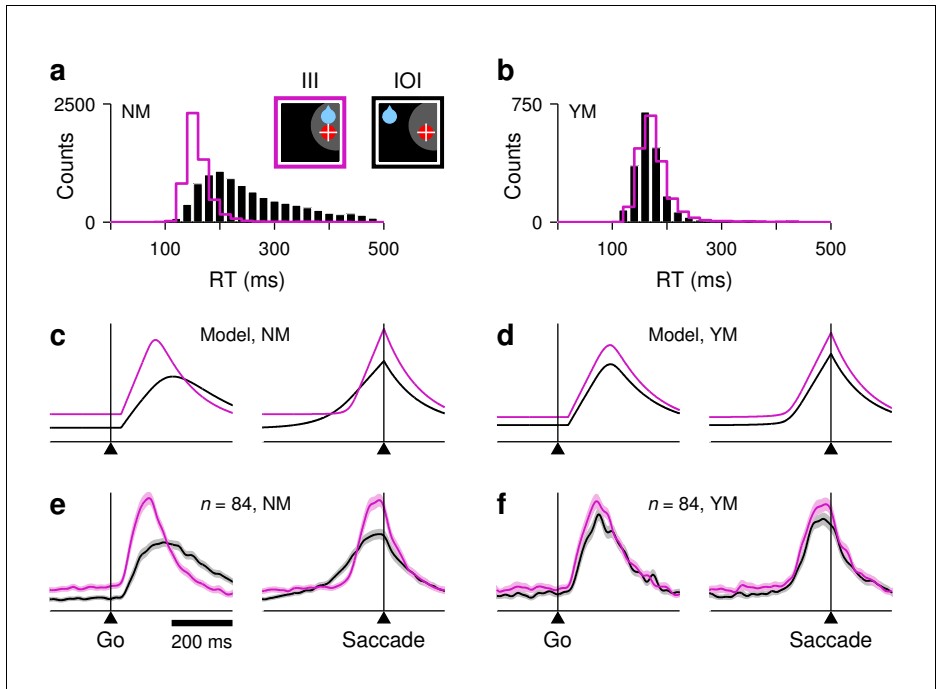

**Figure 6.** Disentangling the coupling between baseline activity, build-up rate, and threshold. (a) Original RT distributions for III (magenta) and IOI (black) trials from all FEF recording sessions (non-matched condition, NM). (b) RT distributions after RT matching (yes-matched condition, YM). (c), Firing rate as a function of time for the target-driven activity, $R_T$, in simulated III (magenta) and IOI (black) trials; same as red traces in *Figure 5d,e*. RTs are not matched. (d) As (c), but with matched RTs. (e) Normalized firing rate as a function of time for a population of 84 V, VM, and M neurons. RTs are not matched. (f) As (e) but with matched RTs.
DOI: https://doi.org/10.7554/eLife.33456.012

The following figure supplement is available for figure 6:

**Figure supplement 1.** Comparisons between congruent and incongruent conditions, with and without RT equalization, for individual cell types.
DOI: https://doi.org/10.7554/eLife.33456.013

no longer intersect; now, across the two conditions, the differences in baseline and threshold are smaller, and the build-up rates are nearly identical (*Figure 6d*). These results make perfect sense within our modeling framework. First, shorter RTs are associated with higher baseline, higher build-up rate, and higher threshold, but the RT equalization procedure only retains the fastest IOI trials, so naturally the resulting IOI trajectory simultaneously increases its baseline, build-up rate, and threshold. Second, the build-up rate changes the most because it relates to RT most directly (*Figure 4b*). And third, the residual difference between congruent and incongruent conditions, which is exclusively related to the internal bias signal, is consistent with the strong coupling between the threshold and the baseline across trials (*Figure 4a*; *Equation 6*).

Now consider the same analysis but for 84 $\mathrm{V}$, $\mathrm{VM}$, and $\mathrm{M}$ FEF neurons that had sufficient RT-matched trials. Qualitatively, the results are just as expected from the model: when the RTs are not matched, the responses in III and IOI trials differ patently in baseline, threshold, and build-up rate (*Figure 6e*), and the corresponding curves intersect each other twice when the spikes are aligned on saccade onset (right panel). In contrast, when the RTs are matched, the differences in build-up rate practically disappear, and those in baseline and threshold become smaller (*Figure 6f*). These results were qualitatively similar across FEF cell categories (*Figure 6—figure supplement 1*).

These findings are consistent with the dynamics of the model, which dictate that shorter RTs are generally associated with a higher baseline (at the target location), higher build-up rate, and higher threshold, where the correlation between build-up rate and RT is strongest.

## Predicted RT sensitivity of the mean population activity

To test the model more stringently, we exploited the wide range of RTs produced in the 1DR task to generate predictions for how the evoked neural activity should be expected to vary as a function of RT. The rationale for these predictions is straightforward: instead of calculating the mean activity averaged across all trials in, say, the IOI condition (*Figure 5e*), we consider similar traces based on subsets of trials within narrow RT bins ('Materials and methods, *Continuous firing activity*'). Assuming that the activity in FEF directly contributes to triggering each saccade, as happens in the model, the resulting response profiles should vary systematically across those RT bins, and any patterns should be consistent with the correlations among baseline, build-up rate, and threshold instantiated by the model, as well as with its other mechanisms (e.g., $R_D$ suppression).

Indeed, when the simulated IOI trials are sorted and averaged according to RT and the resulting curves are color-coded, a characteristic pattern emerges (*Figure 7a*): steeply rising trajectories precede short-latency saccades (black), and more shallow, protracted trajectories precede long-latency saccades (red). This is largely because, in the model, the RT depends critically on the build-up rate of the target-driven activity. Notably, the apparent threshold reached by these curves at saccade onset is also modulated by RT (*Figure 7a*, right panel), with fast choices (black) leading to higher activity levels than slow ones (red). This, in turn, is consistent with the correlated fluctuations in build-up rate and threshold built into the model. Qualitatively, this prediction for the IOI condition — that is, the pattern resulting from the simultaneous dependencies of build-up rate and threshold on RT — is highly robust to parameter variations (of at least ±30%, *Equations 5–11*).

To test this prediction, the recorded trials from 84 FEF neurons were sorted by RT in the same way as the simulated trials, and the corresponding traces were averaged across cells ('Materials and methods, *Continuous firing activity*'). The population curves that resulted (*Figure 7e*) showed the same smooth transitions across RT bins as the simulated curves. Both the build-up rate and the threshold increased with shorter RTs as expected from the model. More generally, when comparing across narrow RT bins, the agreement between the simulations and the overall population activity in FEF was always tight and evident — even though the model predictions varied widely across trial types. This was true in three important respects.

(1) For the activity evoked in III trials. When the target and rewarded locations were congruent, the neural responses into the RF (*Figure 7g*) were much less sensitive to RT than those in the corresponding incongruent trials (*Figure 7e*), with the variations in threshold essentially disappearing (*Figure 7g*, right panel). According to the model (*Figure 7c*), this much weaker dependence resulted from intrinsic randomness, or noise, in the build-up rate, that is, variability that is independent of the baseline (the term $\eta$ in *Equations 8 and 9*). In the model, such randomness is proportionally stronger in III than in IOI trials, and it blurs the effects created by the baseline-dependent fluctuations. This is explained in more detail in the last section of the 'Results'. Here, we simply emphasize that, although

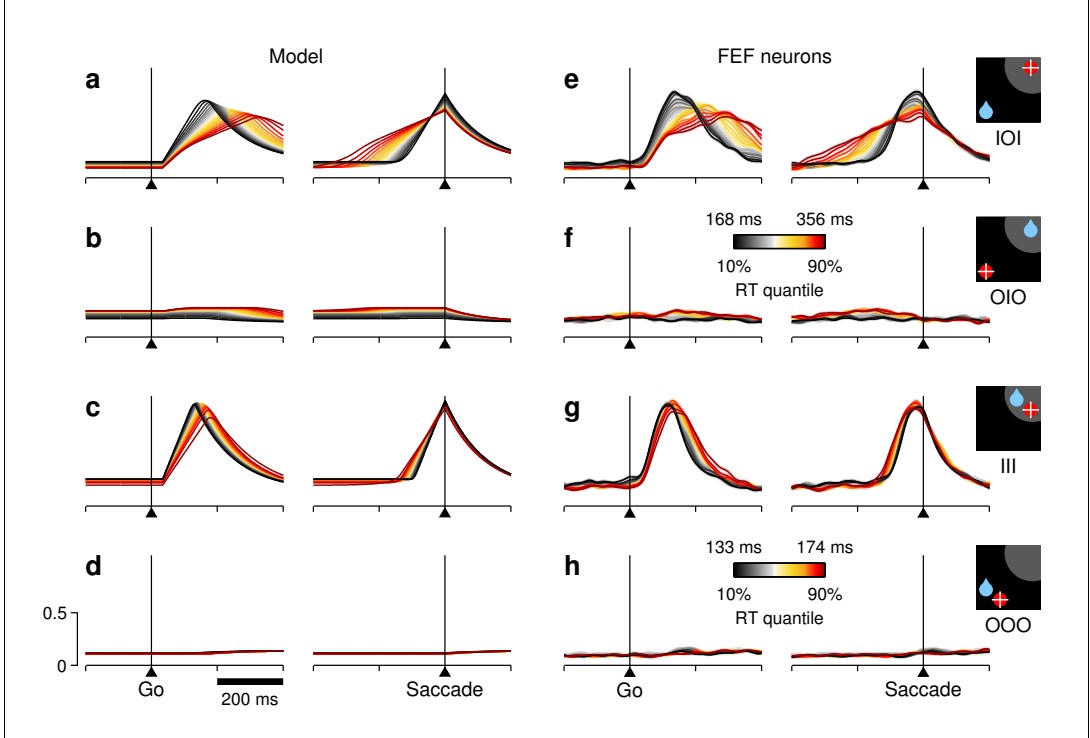

**Figure 7.** RT sensitivity of the average population activity. (a–d) Firing rate as a function of time for the simulated motor plans $R_T$ (a, c) and $R_D$ (b, d) during incongruent (a, b) and congruent (c, d) trials. The corresponding experimental conditions are indicated by the icons (far right). Curves are based on the same simulations as in *Figure 5d–f*. Each colored trace includes 20% of the simulated trials around a particular RT quantile. (e–h) Normalized firing rate as a function of time for a population of 84 FEF neurons (V, VM, and M). Activity is for correct saccades in the four experimental conditions indicated. Each colored trace includes 20% of the trials recorded from each participating cell around a particular RT quantile. Lighter shades behind lines indicate ± 1 SE across cells. Color bars apply to both simulated and recorded data in incongruent (color bar in [f]) or congruent conditions (color bar in [h]). The scale bars in (d) apply to all panels.

DOI: https://doi.org/10.7554/eLife.33456.014

The following figure supplements are available for figure 7:

**Figure supplement 1.** Correlation between RT and activity during correct saccades away from the RF.

DOI: https://doi.org/10.7554/eLife.33456.015

**Figure supplement 2.** RT sensitivity of the VM/M population activity.

DOI: https://doi.org/10.7554/eLife.33456.016

**Figure supplement 3.** Responses in the ADR task, in which all correct trials were equally rewarded.

DOI: https://doi.org/10.7554/eLife.33456.017

the sensitivity to RT manifested quite differently in III and IOI trials, there was close agreement between the simulated and neural data in both.

(2) For the baseline activity. The model postulates that the fluctuations in baseline translate into fluctuations in build-up rate and threshold. As a consequence, when sorted by RT, the simulated baseline levels spread accordingly. For the target-driven response, $R_T$, a higher baseline always corresponds to shorter RTs (*Figure 7a,c*, left panels, note black lines above red before the go signal), so the correlation $\rho(R_b, \mathrm{RT})$ is negative. By contrast, for the baseline of the opposite motor plan, $R_D$, the correlation is either positive (*Figure 7b*, note red lines above black) or zero (*Figure 7d*, note overlapping red and black curves), because of the competitive nature of the interactions between the $T$ and $D$ motor plans. Again, even though the effects were expected to be small for all conditions, the actual baseline activity measured in FEF was highly consistent with the model predictions (*Figure 7e–h*, left panels). This is easier to visualize when the data are magnified appropriately (*Figure 8a*). In quantitative terms (*Figure 8b*), on average, there was a significant negative correlation between baseline activity and RT in both IOI ($p = 10^{-5}$, signed-rank test) and III ($p = 0.004$) trials,

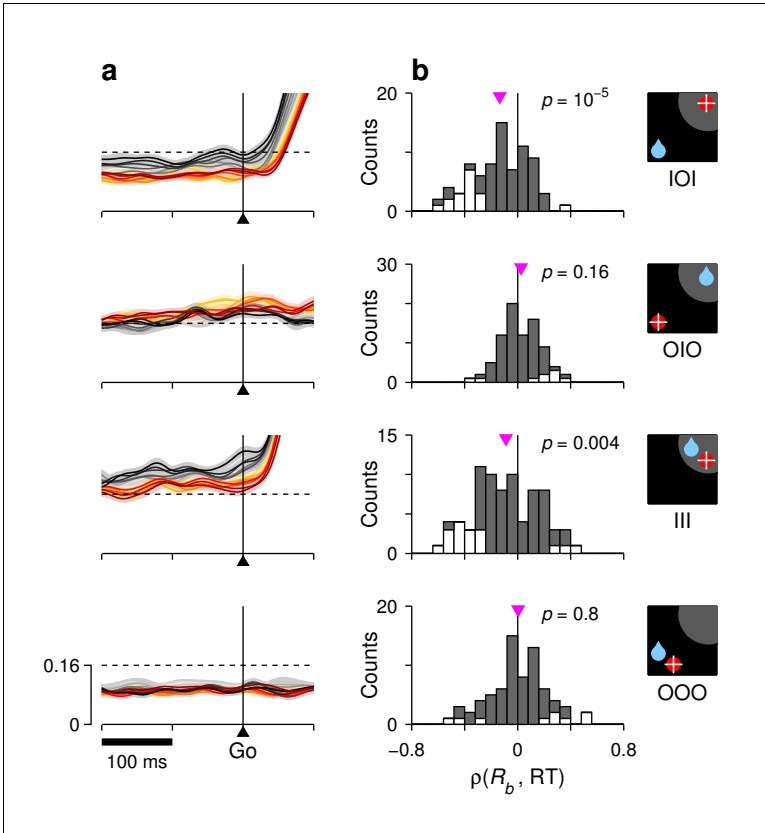

**Figure 8.** Correlation between RT and baseline activity in FEF. For each row, the corresponding target-reward-saccade configuration is depicted by the icon on the far right. (a) Normalized neural responses around the go signal. Each colored trace corresponds to population firing rate as a function of time for a subset of trials around a particular RT quantile. Traces are exactly as in *Figure 7e–h* (left panels), except at shorter x and y scales, as marked on the bottom panel. The dashed reference line is identical across panels. (b) Distributions of Spearman correlation coefficients between RT and baseline activity, $\rho(R_b, \mathrm{RT})$. The data in each histogram are from the same cells used in *Figure 7*. White bars correspond to significant correlations ($p<0.05$). Pink triangles mark mean values, with significance from signed-rank tests indicated.

DOI: https://doi.org/10.7554/eLife.33456.018

as predicted. There was no net correlation in OOO trials ($p = 0.8$), again as predicted. In OIO trials, although the trend was not strong ($p = 0.16$), it was toward a positive correlation, as expected.

(3) For the activity elicited during saccades away from the RF. The model predicts that, when the target is outside of the RF, the low-intensity evoked response should display the same dependencies on RT as the baseline activity preceding it (*Figure 7b,d*). Once again, the neural data were very similar to the model simulations (*Figure 7f,h*; *Figure 7—figure supplement 1*), and the agreement was confirmed statistically (*Figure 7—figure supplement 1*).

In summary, the FEF activity averaged across V, VM, and M cells demonstrated varying degrees of sensitivity to RT, depending on the specific experimental condition considered, but in all cases, the neuronal data conformed closely to the simulation results. Such agreement supports several key features of the model, including the competitive interactions between the target- and bias-driven responses, the limited yet visible impact of intrinsic randomness (in build-up rate) on the evoked responses, and a central hypothesis of the model — that the fluctuations in baseline at the two relevant locations are directly linked and possibly causal to the subsequent movement-related dynamics and, ultimately, to the RTs generated.

## Model robustness

We examined two aspects of the model that could potentially limit its significance. The first is the degree to which it depends on the precise mix of $V$, $VM$, and $M$ cells included in the analyses. This is an issue because it is unclear whether all cell types contribute equally to the FEF output, that is, to the signal that is thought to cross a threshold to trigger a saccade. For example, the purely visual ($V$) neurons could have a lesser weight than those with movement-related activity.

To investigate this, we repeated all analyses excluding all of the $V$ neurons ($n = 26$ cells with visuo-motor index < 0.46) from the population averages, and re-fitted the model in accordance to this more restricted data set. Quantitatively, the main difference was that variations in baseline were diminished, but qualitatively, the results were similar to those based on the larger population (compare *Figure 7* and *Figure 7—figure supplement 2*). Furthermore, the model was still able to simultaneously replicate the neural activity and the RT distributions accurately (*Figure 7—figure supplement 2*). Thus, neither the neuronal averages nor the model results are overly sensitive to the magnitude of the visual component of the population response.

The second potential concern is whether the model generalizes to other tasks or experimental conditions. Consistency with prior studies indicates that the reward manipulation in the 1DR task simply exaggerates and exposes mechanisms that are always operating (see 'Discussion'). However, to test this more directly, we explored the conditions needed for the model to reproduce the data in the ADR task, in which all directions are equally rewarded (*Figure 1b*).

In the ADR condition, the average baseline level was constant (*Figure 7—figure supplement 3*, compare [c] vs. [d]), as expected given that there was no spatial asymmetry in that case. Accordingly, in the model, the two mean baseline levels were set equal to the measured experimental value ($B_T = B_D = 0.2$ in *Equation 5*). Apart from that, matching the model to the ADR data required only two additional parameter adjustments: lowering the variability of the baselines, and slightly decreasing the maximum build-up rate of the stimulus-driven response (*Figure 7—figure supplement 3*). Everything else was as in the 1DR simulations. With those changes, the model generated less extreme variations in baseline, build-up rate, and threshold, and was able to replicate both the neural activity and the RT distribution measured in ADR trials (*Figure 7—figure supplement 3*). These ADR results provide an important consistency check, showing that the same competitive dynamics postulated for the congruent and incongruent 1DR conditions are fully compatible with the simpler, unbiased case.

## Heterogeneity in RT preference across FEF cells

In characterizing the functional roles of specific brain circuits, one of the main challenges is dealing with the inevitable heterogeneity of cell types and their specializations (*Zeng and Sanes, 2017*). Not surprisingly, single FEF neurons showed a variety of relationships to RT in the 1DR task. Remarkably, however, the model accounted for much of this diversity on the basis of a simple intuition: the build-up rate of the target-driven activity is determined by two factors, one that is coupled to the baseline and another that is not, and those factors are weighted in different proportions across single neurons. In this section. we first describe the range of RT preferences measured in single FEF cells and then show that those diverse preferences naturally fall out of the elements already built into the model.

We examined the responses of individual FEF neurons during correct saccades into the RF, and found that their dependencies on RT could deviate quite substantially from those of the average population. This is illustrated with two example cells for which the maximum level of activity across trials was modulated strongly — and in opposite directions (*Figure 9a–f*). To view all the responses recorded from a given neuron simultaneously, these were arranged as activity maps in which color corresponds to firing intensity and trials are ordered according to RT (*Figure 9a,d*). In this way, it is clear that both cells were most active shortly before the saccade (white marks on the right) and that their firing rates were very different for the fastest versus the slowest responses (top vs. bottom trials). One cell preferred fast trials, that is, it fired at higher rates for short RTs, whereas the other preferred slow trials, that is, it fired at higher rates for long RTs. The contrast is also apparent when the same data are plotted as collections of firing rate traces sorted and color-coded by RT, as done in previous figures (*Figure 9b,e*; compare to *Figure 7e*).

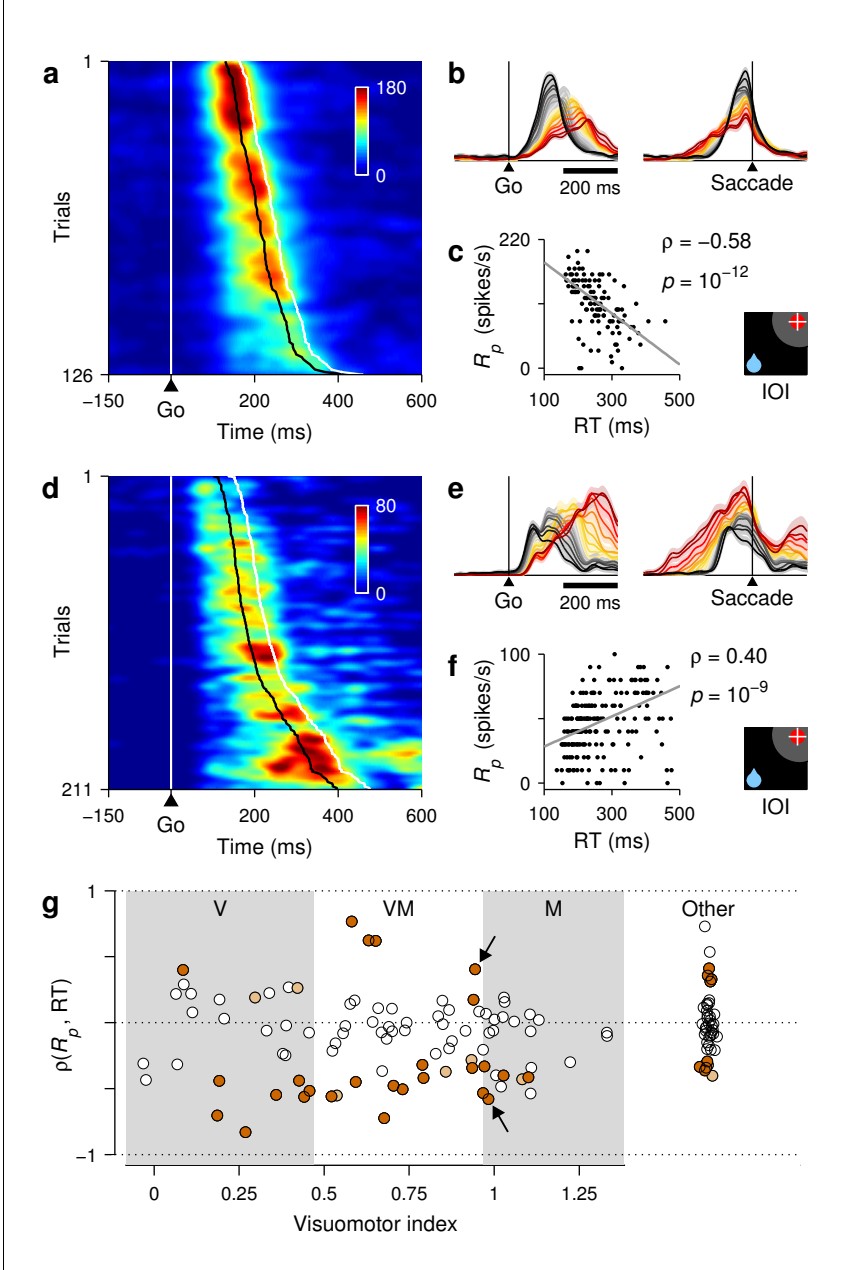

**Figure 9.** RT sensitivity of individual FEF neurons. (a–c) Responses of a single FEF cell that fired preferentially during short RTs. All data are from IOI trials (icon). (a) Activity map. Each row corresponds to one trial, and shows the cell's firing rate (represented by color) as a function of time (x axis) with spikes aligned to the go signal (vertical white line). Trials are ordered by RT from fastest (top) to slowest (bottom). For this cell, the time of peak activity (black marks) closely tracks saccade onset (white marks). (b) Firing rate as a function of time, with trials sorted and color-coded by RT, as in *Figure 7*. (c) Peak response as a function of RT. Each dot corresponds to one trial. The gray line shows linear fit. Numbers indicate Spearman correlation $\rho(R_p, \text{RT})$ and significance. (d–f) As (a–c) but for a single cell that fired preferentially during long RTs. (g) Spearman correlation between peak response and RT for all classified cells ($n = 132$). For standard cell types (V, VM, M), values on the abscissa correspond to visuomotor index; for other cells, they are arbitrary. Shades demarcate ranges approximately corresponding to standard V, VM, and M categories. All correlation values are from IOI trials. Brown points indicate significant neurons (light, $p < 0.05$; dark $p < 0.01$). Arrows identify example cells in panels above.
DOI: https://doi.org/10.7554/eLife.33456.019

For each recorded neuron, sensitivity to RT was quantified by $\rho(R_p, \mathrm{RT})$, the Spearman correlation between the peak response and RT across trials ('Materials and methods, *Statistical analyses*'). Negative values correspond to a preference for short RTs, as for the first example cell (*Figure 9c*), whereas positive values correspond to a preference for long RTs, as for the second example cell (*Figure 9f*). Across our FEF sample, the resulting distribution of correlation coefficients (mean $\rho = -0.12$, $p = 0.0005$, signed rank test) was notable in two ways. First, it contained many more significant correlations, both positive and negative, than expected just by chance (43 of 132 cells $\sim 33\%$ were significant with $p < 0.05$, as opposed to 6.75 expected by chance; $p = 10^{-22}$, binomial test). Thus, a substantial fraction of the FEF neurons had robust temporal preferences, with both modulation types represented (*Figure 9g*, colored points). Second, the distribution was approximately the same for all the standard FEF cell types. The proportion of positive and negative correlations, as well as the fraction of significant neurons, was statistically the same for the V, VM, M, and other categories (*Figure 9g*; $p > 0.2$, binomial tests). So, as far as we could tell, the sensitivity to RT spanned a similar range for all the elements of the FEF circuitry.

These results explain the moderate RT sensitivity seen in the average population activity during IOI trials (*Figure 7e*) as the sum of two opposing contributions from subpopulations with temporal preferences that partially offset each other. Within the fast-preferring group ($\rho < 0$), the pattern of response trajectories of many cells was qualitatively similar to that of the average population but showed more extreme variations across RT bins (*Figure 10f*; compare to *Figure 7e*). By contrast, only a few of the neurons within the slow-preferring group ($\rho > 0$) demonstrated a strong correlation with RT (*Figures 9d–f* and *10g*); for most of them, the dependence on RT, particularly during the $\sim 100$ ms before movement onset, was more modest (*Figure 10h*). Thus, when the responses of the fast- and slow-preferring neurons are combined, their temporal dependencies partially cancel out, and the overall population activity ends up resembling an attenuated version of the fast-preferring responses. Similar results were obtained in the congruent condition. That is, both fast- and slow-preferring cells were also found during III trials (*Figure 10i,j*), except that in this case, the complementary modulations canceled out more fully upon averaging (*Figure 7g*).

The temporal heterogeneity just discussed was readily replicated by the model. For incongruent trials, strong modulation could be simulated favoring either short (*Figure 10a*) or long RTs (*Figure 10b*), but more modest temporal sensitivity like that exhibited by the majority of slow-preferring neurons could be reproduced too (*Figure 10c*). In all of these cases, the simulated response trajectories matched the experimental results extremely well (compare to *Figure 10f–h*). The most visible (but still minor) discrepancy between the simulated and neuronal trajectories was due to the discontinuity of the threshold-crossing event in the former, as opposed to the sharp but smooth turn around the peak of activity of the latter. Analogous results were obtained in congruent trials, for which both fast- (*Figure 10d*) and slow-preferring (*Figure 10e*) model responses similar to the experimental ones were also generated (compare to *Figure 10i,j*). In both congruency conditions, the simulated slow-preferring responses are particularly notable because, although they are still target-driven, they would seem to require mechanisms that are completely different from those described earlier.

How did the model capture such wide-ranging heterogeneity? The short answer is that, without changing any of the parameters, the target-driven activity in the model, $R_T$, can be naturally expressed as the sum of two components:

$$R_T(t) = R_T^U(t) + R_T^C(t) \tag{1}$$

where the equality holds at every point in time. For one component (indicated by the $C$ superscript), the variations in build-up rate are coupled to the fluctuations in baseline ($B_T$), whereas for the other component ($U$ superscript), the variations in build-up rate are random, uncoupled from the baseline. These two components correspond to the two neuronal types with opposite RT preferences.

To see the correspondence, consider once more the RT-sensitive responses of the FEF cells, now focusing on how the response trajectories fan out when they are aligned on the go signal: for the fast-preferring examples, the slopes of the curves increase progressively as the RTs become shorter, and the spread is visible from the moment the activity begins to rise (*Figure 10a,d,f,i*, left panels); by contrast, for the slow-preferring examples, all the curves begin to rise with approximately the same slope, and the modulation by RT begins to manifest only later, $\sim 100$ ms after the go signal

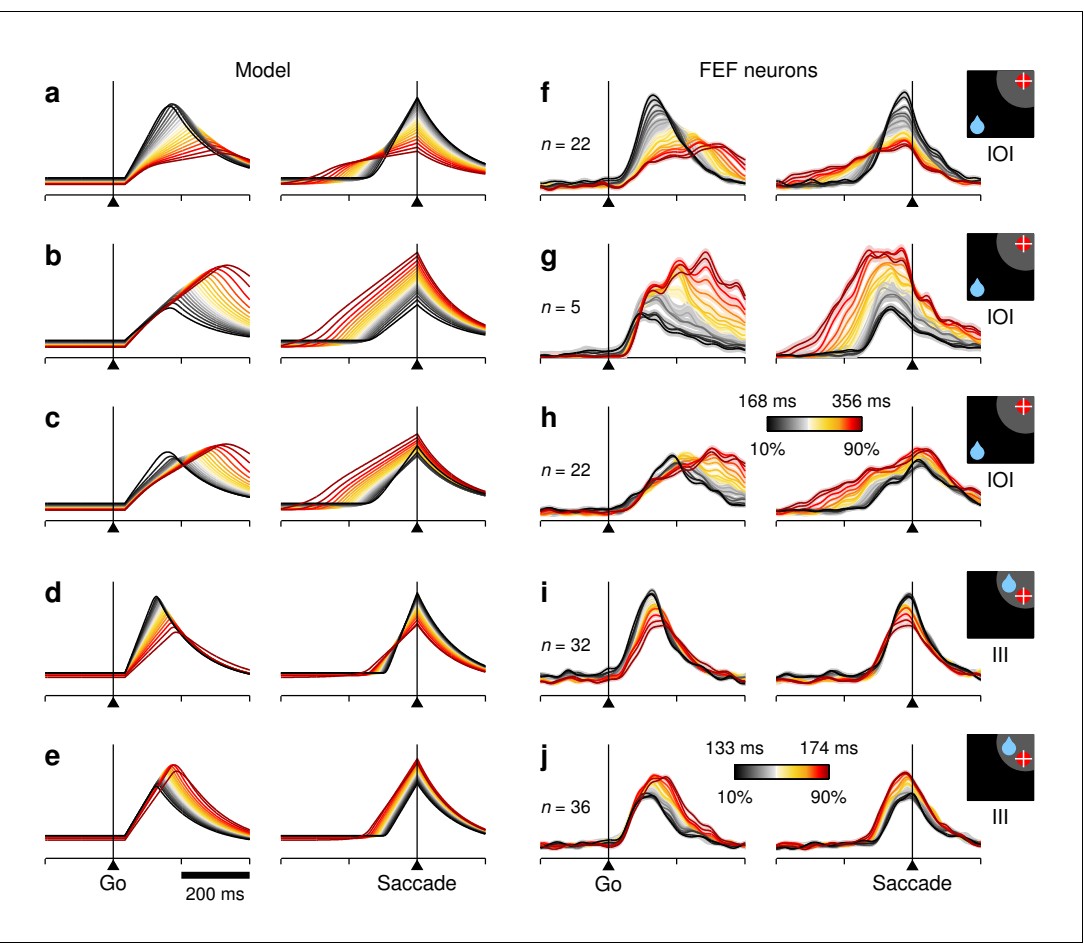

**Figure 10.** RT sensitivity in subpopulations of FEF neurons. All panels show firing rate as a function of time, with each colored curve based on a different subset of trials selected according to RT, as in *Figure 7*. Only activity associated with correct saccades into the RF is displayed. (a–e) Simulated activity demonstrating a clear preference for either short (a, d) or long RTs (b, c, e). The model generated fast- and slow-preferring responses in both IOI (a–c) and III trials (d, e) with varying modulation strengths. (f–j) As in (a–e) but based on the responses of various subsets of FEF neurons with similar RT preferences, as quantified based on $\rho(R_p, \mathrm{RT})$. Each subset is a selection from the same pool of 84 recorded neurons used in *Figure 7* (in [f] $\rho<0$ and $p<0.05$; in [g] $\rho>0$ and $p<0.05$; in [h] $\rho>0$; in (i) $\rho<0$; in [j] $\rho>0$; visuomotor index >0.4 in all cases). Numbers of participating cells are indicated in each panel. Lighter shades behind lines indicate ± 1 SE across cells. Color bars apply to both simulated and recorded data, either in IOI (color bar in h) or III trials (colorbar in j).
DOI: https://doi.org/10.7554/eLife.33456.020

(*Figure 10b,e,g,j*, left panels). According to the model, this feature, the variability of the initial build-up rate, is the fundamental mechanistic distinction between the fast- and slow-preferring FEF neurons.

A more elaborate intuition can be gleaned from the analytical expression that determines the initial build-up rate of the target-driven activity, $G_T$, in each trial. This build-up rate can be written as

$$G_T = f_1 + \phi + f_2 B_T \tag{2}$$

(see *Equations 8, 9*). Here, the terms $f_1$, $f_2$ and $\phi$ are not necessarily constant, but what matters is that they do not depend on the baseline at the target location, $B_T$. The term $\phi$, which for now is assumed to be relatively small, represents noise in $G_T$, that is, the random fluctuations in build-up rate mentioned earlier. Intuitively, then, *Equation 2* says that the initial build-up of $R_T$ is the result of two influences, a term that depends on the baseline $B_T$ and a relatively constant drive that is

independent of it. The former, $f_2 B_T$, leads to much higher variability in build-up rate across trials — and stronger covariance with RT — than the latter, $f_1 + \phi$.

Now, the coupled and uncoupled components in *Equation 1* differ exclusively in their initial build-up rates, which are given by the two terms just discussed:

$$G_T^U = \alpha f_1 + f_1 + \phi$$

$$G_T^C = -\alpha f_1 + f_2 B_T . \tag{3}$$

A key property of these build-up rates is that their sum, $G_T^U + G_T^C$, is always equal to $G_T$, as given by *Equation 2*. This is true for any value of the newly introduced parameter $\alpha$, which serves as an offset by means of which the weights of the two components may be adjusted. Splitting the target-driven activity in this way allowed us to simulate neuronal responses $R_T^U$ and $R_T^C$ that had opposite RT preferences but were otherwise identical, as they had the same initial conditions, afferent delays, evolution equations, and so on. Furthermore, by varying $\alpha$, we could vary the strength of the resulting modulation — without altering either the original target-related activity, $R_T(t)$, or the outcomes and RTs of the competitions in any way. In other words, the split via *Equations 1–3* produces paired sets of target-driven responses with different RT sensitivities, parameterized by $\alpha$, but all the pairs thus generated are compatible with the same summed activity (*Figure 7*) and the same distributions of outcomes and RTs (*Figure 5g–i*). On the basis of this simple decomposition of $R_T$ into pairs of components, the model can generate a wide range of RT-sensitive responses, which are strikingly similar to those found in the FEF population.

In closing this section, we emphasize the distinct role that intrinsic randomness plays in the model, and why it is necessary. During incongruent trials (or, more precisely, when $B_T \leq B_D$). the variance in RT is so strongly coupled to the fluctuations in baseline activity that the noise in the build-up rate has a negligible impact (in *Equation 2*, $\phi \ll f_2$). In that case, even large variations (of ~100%) in noise have relatively little consequence, and $R_T^C$ and $R_T^U$ correspond almost perfectly to the fast- and slow-preferring neurons, respectively. By contrast, during congruent trials (or, more precisely, when $B_T > B_D$), a relatively large amount of independent noise ($\phi > f_2$) is necessary to reproduce the lack of sensitivity to RT in the average population activity (*Figure 7g*). In that case, the preferences for short and long RTs of the simulated responses (*Figure 10d,e*) are noticeably sensitive to modest variations (of ~20%) in noise, and the temporal heterogeneity of the $R_T^U$ and $R_T^C$ components is more complicated; the details are beyond the scope of this report. Nevertheless, the conclusion is clear: the large variance in RT observed during incongruent trials vastly exceeds that associated with intrinsic noise in the rising activity, and is fundamentally determined by the covert conflict between competing saccade plans and the ensuing dynamics; whereas the much smaller variance in RT observed during congruent trials is best explained by noise that is independent of the competitive interactions, which make just a modest contribution in that case.

## Discussion

Reactive saccades are to voluntary behavior what the harmonic oscillator is to classical mechanics — the simplest non-trivial problem. And yet, the fundamental mechanisms that determine their timing have remained poorly understood. We revisited the established account of how saccades are programmed, the rise-to-threshold process, and found new dynamics that explain why the variance of saccadic RTs is high even under minimalistic stimulation conditions. Key to this was the saccadic competition model, which connected numerous empirical observations and provided a deeper understanding of the full RT distributions in terms of circuit interactions.

### Motor conflict is critical

We examined single-neuron activity in FEF, a cortical area whose role in saccade generation and attentional deployment is firmly established (*Bruce and Goldberg, 1985*; *Tehovnik et al., 2000*; *Squire et al., 2012*), and where the activity of movement-related neurons is perhaps most emblematic of the idealized rise to threshold (*Hanes and Schall, 1996*; *Fecteau and Munoz, 2007*; *Stanford et al., 2010*; *Ding and Gold, 2012*; *Costello et al., 2013*). We found that the three main components of this process — the baseline activity preceding target presentation, the build-up rate

of the evoked response, and the saccade threshold — fluctuate in a coordinated fashion. This is already a significant departure from the simpler, standard framework in which the only source of variability (within a given experimental condition) is the build-up rate. But the problem is more complicated, because by themselves, the interrelationships between these three variables are insufficient to explain the variations in RT accurately; for that, it is critical to consider not only the target-driven response but also the weaker, internally driven activity favoring saccades to alternative locations.

During reactive saccades, it may seem as if only one motor plan is possible, but this is rather illusory. When the target appears, oculomotor activity begins to grow in response to it, but this ramping process (represented by $R_T$ in the model) does not start from the same neutral state every time; instead, it occurs while other incipient, internally driven motor plans (represented by $R_D$) are also developing, and the time necessary to resolve the ensuing conflict depends on how advanced and how congruent those budding, bias-driven plans are relative to the target-driven response. Thus, the high variance in RT results not from noisy representations or sloppy computations, but rather from the normal operation of a well-oiled motor-selection machine (*Najemnik and Geisler, 2005*; *Oostwoud Wijdenes et al., 2016*; *Tian et al., 2016*). Making the target unique, highly visible, and task-relevant minimizes potential variance related to the sensory detection step and enhances the priority of the target-related plan, but still leaves those alternative internal plans relatively unconstrained. What the 1DR task does is to align the internal biases with a specific direction — that of the expected reward — and it is under those conditions that the motor selection process becomes more apparent.

Contrary to current ideas, we found that noise in the build-up rate is not the main source of RT variance. Such noise was discernible but only during congruent trials, when the target-driven activity typically starts with the higher baseline and there is minimal motor ambiguity to begin with (*Figure 3a*). More typically, the proportion of variance in RT due to such intrinsic randomness is smaller (ADR task), and may become negligible (1DR task, incongruent trials) as motor conflict increases.

The model revealed completely novel mechanistic details of the motor selection process but, on average, its manifestations during the rise to threshold were nevertheless quite subtle, particularly during correct saccades. Consider, for example, how the longest RTs in correct trials are produced (*Figure 5b,h*). That was quite a puzzle. In that case, the target-driven activity, which starts with a lower baseline than the opposing plan, must somehow rise at about the lowest possible build-up rate and yet still win the competition by a large margin. The solution is for the target-driven response to rise quickly initially, suppress the opposing plan early on, and then slow down immediately afterward — all of which happens when the competing plans are far from threshold. This is a major departure from standard choice models, in which the outcome is not determined until activity is close to or at threshold. It means that critical dynamical interactions may occur at quite low levels of activity, where they are less readily apparent and much more difficult to characterize without prior knowledge of their signature features.

The model has numerous moving parts, but consider the scope of the data that it reconciles ('Materials and methods, *Correspondence between data and model parameters*'). Behaviorally, the model generated errors at the appropriate rates and reproduced three distinct RT distributions in their entirety (*Figure 5g–i*). Neurophysiologically, it replicated the average response trajectories in all conditions (*Figure 5d–f*), the isolated effect of the spatial bias (*Figure 6c–f*), the dependence on RT of the average activity (*Figure 7*), and most remarkably, the responses of individual FEF neurons, which showed a wide range of RT preferences (*Figure 10*). With minimal adjustment, the model also replicated the empirical results in the ADR task (*Figure 7—figure supplement 3*). Thus, a large number of disparate observations are subsumed into one coherent framework for resolving the kind of saccadic conflict that must typify naturally occurring oculomotor behaviors.

## The threshold varies — but not as we thought

Our results indicate that the saccade threshold is not constant. Instead, it fluctuates quite dramatically, and in tandem with other elements of the circuitry. Although the proposed dynamics remain to be tested directly, substantial agreement can already be found with extant data. For instance, in the model, a stronger internally driven activity promotes a lower threshold, and indeed, movement-related activity preceding memory-guided saccades, anti-saccades, or saccades triggered by blinks is considerably weaker than that for stimulus-driven saccades (*Edelman and Goldberg, 2001*;

*Jantz et al., 2013*; *Jagadisan and Gandhi, 2017*). In addition, the presaccadic activity measured during visual search is less vigorous for incorrect than for correct responses to the same location (*Thompson et al., 2005*), presumably because the former involve a stronger internal (and erroneous) drive that promotes a lower threshold, just as in our data. And, again in the context of visual search, one study (*Heitz and Schall, 2012*) showed multiple differences in FEF activity across task conditions similar to those found here; that is, when the mean RT was shorter, the baseline, build-up rate, and threshold were all higher.

These findings place significant constraints on the trigger mechanism that converts a saccade plan into a committed, uncancelable command. For instance, a popular idea is that adjustments in threshold may serve to trade speed against accuracy during choices (*Lo and Wang, 2006*; *Bogacz et al., 2010*; *Heitz and Schall, 2012*; but see *Salinas et al., 2014*; *Thura et al., 2014*; *Thura and Cisek, 2016*, *2017*). This is simply because, everything else being equal, activity that ramps toward a higher threshold should take longer to reach it, thus providing more time for deliberation. However, the results in the 1DR task are entirely antithetical to these notions: first, congruent trials produce shorter RTs and *higher* accuracy than incongruent ones, and second, the observed variations in threshold are linked to variations in baseline and build-up rate in ways that, according to the standard characterization of the speed-accuracy tradeoff, are entirely inconsistent. A higher threshold, which lengthens the deliberation period, is typically accompanied by a higher baseline and a higher build-up rate, both of which shorten that period.

These considerations are significant because, although the threshold is a universal feature of decision-making models, the strongest evidence of its existence is precisely the behavior of oculomotor neurons in FEF and SC (*Hanes and Schall, 1996*; *Lo and Wang, 2006*; *Brown et al., 2008*; *Stanford et al., 2010*; *Ding and Gold, 2012*). In other related circuits, either no threshold is apparent (*Ding and Gold, 2010*; *Stuphorn et al., 2010*) or its implementation is much less evident (*Afshar et al., 2011*; *Hayden et al., 2011*).

## It is all about the base

*Sumner (2011)* has pinpointed why explaining saccadic latency distributions has been so challenging: it is well established that Gaussian variability in the build-up rate of a rise-to-threshold process accurately reproduces their characteristic skewed shapes (*Carpenter and Williams, 1995*; *Hanes and Schall, 1996*; *Fecteau and Munoz, 2007*), but most factors that are known to affect saccadic RT in simple tasks are normally modeled as baseline shifts (*Sumner, 2011*). Our results suggest that the dichotomy is false. The fluctuations in baseline, build-up rate, and RT are inextricably linked (*Figures 4*, *6*, *7* and *8*); they involve alternate, covert motor plans and depend on multiple neural mechanisms acting in concert.

According to the model, the major source of randomness across trials is the variability of the baselines (*Equation 5*). The noise in the target-driven build-up rate ($\phi$ in *Equation 2*) makes a distinct contribution, but in the incongruent condition, in particular, nearly all of the variance observed experimentally — in RT, in saccadic choice, in threshold level, and in the build-up rates and peak responses of the neurons — is determined by the computational amplification of the initial baseline fluctuations.

This is a simplification, of course. First, it is possible that the link is not strictly causal, that is, that an unidentified factor drives the fluctuations in baseline and in the other dynamic variables. And second, other sources of variability are likely to contribute too; for instance, variations in response onset (*Pouget et al., 2011*; *Peel et al., 2017*), which would correspond to fluctuations in the afferent delays of the model. These contributions, however, are likely to be very small in comparison (*Pouget et al., 2011*). It is certainly possible to add noise to the afferent delays or to other components of the model without substantially altering the results, but what is notable is that this is not necessary.

Arguably, the baselines reflect multiple cognitive elements, including expectation, anticipation, and the allocation of attention and other resources (*Bruce and Goldberg, 1985*; *Kastner et al., 1999*; *Coe et al., 2002*; *Maunsell, 2004*; *Rao et al., 2012*; *Zhang et al., 2014*; *Thura and Cisek, 2016*). In the model, these elements set the initial spatial priorities such that the neurons with the higher baseline have a higher probability of triggering the next saccade. This probability is a complicated function of the baseline values and the upcoming stimulus, but qualitatively, the effect is very much in agreement with the findings of other single-neuron studies (*Everling et al., 1998*;

*Everling and Munoz, 2000*) and with the results of subthreshold microstimulation experiments (*Glimcher and Sparks, 1993*; *Dorris et al., 2007*). It is also in line with theoretical studies showing that the background activity in recurrent circuits can have profound dynamical and amplification effects on evoked responses (*Salinas and Abbott, 1996*; *Chance et al., 2002*; *Salinas, 2003*; *York and van Rossum, 2009*).

The model proposes a tight relationship between initial state and subsequent oculomotor dynamics, and interestingly, mounting evidence demonstrates a similar phenomenon in motor cortex, whereby the initial neural state is predictive of an ensuing arm movement and of the trajectories of the underlying neural signals over time (*Churchland et al., 2006*, *2010*, *2012*; *Afshar et al., 2011*; *Ames et al., 2014*; *Sheahan et al., 2016*; *Stavisky et al., 2017*; *Wang et al., 2018*). In that case, the dynamics develop within a very high-dimensional space, such that the preparatory activity is only weakly related to specific kinematic parameters (*Churchland et al., 2006*, *2010*). Saccades are simpler because they are lower dimensional and largely stereotyped, and because the activity of any given cell generally corresponds to a fixed movement vector. However, we propose that their dynamical behavior is qualitatively similar in that the initial state of the system — that is, the configuration of baseline levels across RFs during fixation — fundamentally determines its subsequent temporal evolution, including its interaction with new incoming sensory information (*Sheahan et al., 2016*) and the eventual outcome (*Churchland et al., 2006*; *Afshar et al., 2011*).

Overall, the current results suggest that, when oculomotor circuits receive new visual information, ongoing saccade plans and internal settings (e.g., threshold level and attentional locus) radically shape the impact of that information, even when it is behaviorally relevant, expected, and unambiguous. Deeper understanding of the underlying network dynamics will be critical for further elucidation of how saccades are triggered and, more generally, of how perceptually guided choices are made.

## Materials and methods

### Subjects and setup

Experimental subjects were two adult male rhesus monkeys (*Macaca mulatta*). All experimental procedures were conducted in accordance with the NIH Guide for the Care and Use of Laboratory Animals, USDA regulations, and the policies set forth by the Institutional Animal Care and Use Committee (IACUC) of Wake Forest School of Medicine.

An MRI-compatible post served to stabilize the head during behavioral training and recording sessions. Analog eye position signals were collected using a scleral search coil (Riverbend Instruments, Birmingham, AL) and infrared tracking (EyeLink 1000, SR Research, Ottawa, Ontario, Canada). Stimulus presentation, reward delivery, and data acquisition were controlled by a purpose-designed software/hardware package (Ryklin Software, New York, NY). Target stimuli were displayed by a $48 \times 42$ array of tri-color light-emitting diodes. Saccade onset was identified as the time at which eye velocity exceeded 50°/s; having detected the start of a saccade, its end was identified as the time at which eye velocity fell below 40°/s. Eye movements were scored as correct if the saccade endpoint fell within 5° of the target stimulus.

Neural activity was recorded using single tungsten microelectrodes (2–4 M$\Omega$, FHC, Bowdoin, ME) driven by a hydraulic microdrive (FHC). Individual neurons were isolated on the basis of the amplitude and/or waveform characteristics of the recorded and filtered signals (FHC; Plexon, Inc, Dallas, TX). Putative FEF neurons were selected from areas in which saccade-like movements could be evoked by low-current microstimulation (70 ms stimulus trains at 350 Hz, with amplitude equal to 50 $\mu$A) (*Bruce and Goldberg, 1985*; *Costello et al., 2013*). Neurons were recorded unilaterally in both monkeys. The majority of RFs (76%) were located at 10° of eccentricity.

### Behavioral tasks

In the 1DR task (*Figure 1a*), all trials began with the appearance of a centrally located stimulus. Monkeys had to maintain their gaze on this fixation spot for 1000 ms. The disappearance of the fixation spot (go signal) was simultaneous with the appearance of a second, target stimulus in one of four possible locations (*Figure 1a*). Subjects were required to make a saccade to the peripheral target within 500 ms of the go signal in order to receive a liquid reward. Target locations varied

pseudorandomly from trial to trial. In each block of trials, only one of the four target locations was associated with a large reward; the other three were unrewarded (Monkey G), or yielded a much smaller reward (Monkey K). For brevity, we refer to these simply as the 'rewarded' and 'unrewarded' locations. The rewarded location changed pseudorandomly from one block to another. Block length was highly variable (range: 10–140 trials); the average was 70 trials per block.

In the all-directions-rewarded task (ADR), the events were the same as in the 1DR, but the four target locations were rewarded equally (*Figure 1b*). Blocks of ADR trials were run sporadically, interleaved with those of 1DR trials.

In the delayed-saccade task, each trial began with fixation of a central spot, followed by the appearance of a single stimulus at a peripheral location during continued fixation. After a variable delay (500, 750, or 1000 ms), the fixation spot was extinguished (go signal) and the subject received a liquid reward if a saccade was made to the peripheral target. In each experimental session, the delayed-saccade task was run first to locate the RF of the recorded neuron, and subjects performed the 1DR task after the initial spatial characterization.

The four target locations in the 1DR task were chosen on the basis of the RF of each recorded neuron. One location always corresponded to the RF center. The others had equal eccentricity and were 90°, 180°, and 270° away from the RF relative to fixation.

The RT was always measured from the go signal until the onset of the saccade.

## Trial selection

For all analyses not specifically examining sequential effects and block transitions, we discarded the first 8 trials of each 1DR block, during which the monkeys may have been discovering the new rewarded location (*Figure 2—figure supplement 1a*). This guaranteed that all behavioral and neural metrics reflected a stable expectation, and that erroneous saccades were not due to spatial uncertainty. More stringent exclusion criteria produced qualitatively similar results.

## Continuous firing activity

Continuous (or instantaneous) firing rate traces, also known as spike density functions, were computed by convolving evoked spike trains with a Gaussian function ($\sigma = 15$ ms) with unit area. Continuous mean traces for each neuron were generated by averaging across trials. To produce equivalent population responses (e.g., *Figure 3a–c*), the continuous traces of individual cells were first normalized by each neuron's overall maximum firing rate and were then averaged across neurons.

To visualize how RT modulated the activity of each cell, families of firing rate traces ordered and color-coded by RT were generated (*Figure 9b,e*). For this, the trials in the relevant experimental condition (e.g., IOI, III) were sorted by RT and distributed over 20 evenly spaced, overlapping bins defined by RT quantiles, where each bin contained 20% of the trials. Thus, the first bin was centered on the 10th percentile and included the fastest 20% of the recorded trials, the next bin included the 20% of the trials around the 14th percentile, and so on, with the last bin being centered on the 90th percentile and including the slowest 20% of the recorded trials. Then, a continuous firing rate trace was produced for each of the 20 RT bins/quantiles. To generate equivalent families of curves for populations of neurons (*Figure 7e–h*), the traces for each participating cell were first normalized by that cell's overall maximum firing rate, and then, for each quantile, a population trace was compiled by averaging across neurons. This method — based on quantiles, as opposed to standard binning using fixed RT values — reveals more clearly the actual modulation range of the neural responses and their smooth dependence on RT, because the numbers of trials and neurons remain nearly constant across bins. Also, if the dependence on RT is monotonic, as the data indicate, this procedure can only *under*estimate the magnitude of the modulation. For the simulated data, families of response curves ordered by RT were generated in the same way.

For each neuron, an activity map (*Figure 9a,d*) for a given condition (e.g., IOI, III) was assembled by aligning all spike trains to the go signal, converting each one to a continuous firing rate, sorting the trials by RT, putting the sorted firing rate traces into a single matrix, and displaying the matrix as a heat map with color indicating intensity. For display purposes, activity maps were also smoothed with a Gaussian function in the vertical direction, that is, across trials ($\sigma = 2$ trials), but this was exclusively for ease of viewing; trials were kept independent in all analyses.

## Statistical analyses

All data analyses were performed using customized scripts in MATLAB (The MathWorks, Natick, MA). For comparisons across any two conditions, significance was typically evaluated using permutation tests for paired or unpaired samples (*Siegel and Castellan, 1988*), as appropriate. Because 100,000 permutations were used, the smallest significance value in this case is reported as $p < 10^{-5}$.

The relationship between neuronal activity and RT for each neuron was evaluated separately for each experimental condition (e.g., III, IOI) using the Spearman rank correlation coefficient. The MATLAB function *corr* was used for this. This coefficient serves to identify any monotonic relationship between two variables. As measures of activity, for each cell we considered: the baseline response, $R_b$ (firing rate in a 250 ms window preceding the go signal); the mean response, $R_m$ (firing rate computed over the full RT interval); the presaccadic response, $R_{sac}$ (firing rate in a 50–70 ms window preceding saccade onset); the build-up rate, $R_{BU}$ (described below); and the peak response, $R_p$ (described below). The correlation between $R_p$ and RT is denoted as $\rho(R_p, \mathrm{RT})$. Correlations in other activity measures are denoted similarly). Neurons with $\rho(R_p, \mathrm{RT}) < 0$ and $\rho(R_p, \mathrm{RT}) > 0$ were designated as fast- and slow-preferring, respectively.

## Peak response

The standard presaccadic firing rate, $R_{sac}$, which is calculated in a fixed time window anchored to saccade onset, would seem to be the most direct indicator of saccade threshold. However, for a purely visual neuron responding to target onset, the use of $R_{sac}$ could produce a negative correlation with RT even if that cell was always activated with the same temporal profile. To avoid this spurious correlation due to temporal misalignment between the visually driven spikes and saccade onset across trials, we computed the peak response, $R_p$, which is insensitive to the alignment of the spike trains. Although, on average, results based on $R_p$ and $R_{sac}$ were highly consistent (e.g., compare *Figure 3d* and *Figure 5—figure supplement 1a*), the former is a more veridical indicator of response modulation on a single-cell basis.

For each neuron, the value of $R_p$ in each trial was equal to the cell's firing rate computed in an interval centered on the time point $T_p$, which we call the time of peak response. This is simply the time along a trial (with the go signal at $t = 0$) at which the cell was most likely to fire at the highest rate (*Figure 9a,d*, black marks). $T_p$ is described as a linear function of RT,

$$T_p = \beta_0 + \beta_1 \times \mathrm{RT} \tag{4}$$

where the coefficients $\beta_0$ and $\beta_1$ characterize the timing of each neuron. For example, for a typical visual cell with $\beta_0 = 80$ and $\beta_1 = 0$, the maximum rate in a trial is observed 80 ms after the go signal, regardless of RT, whereas for a typical movement-related cell with $\beta_0 = -30$ and $\beta_1 = 1$, the highest discharge occurs 30 ms before the saccade.

The coefficients $\beta_0$ and $\beta_1$ were obtained in two steps: (1) finding, from the activity map of the cell, the maximum instantaneous firing rate in each trial and the time, relative to the go signal, at which that rate was achieved, $T_{max}$, and (2) fitting $T_{max}$ as a linear function of RT. All trials in which a saccade was made into the cell's RF were included, regardless of bias condition. The coefficients resulting from the fit, that is, the intercept and slope, were $\beta_0$ and $\beta_1$. Finally, to determine $R_p$ in a given trial, first, the corresponding $T_p$ was found by plugging the RT from that trial into *Equation 4*; then a firing rate was calculated by counting the spikes in a time window centered on $T_p$ and dividing this by the window length. The result was $R_p$. The window length was 100 ms for most cells (~80%) but was set to 200 ms for a minority that had more prolonged responses (e.g., postsaccadic cells).

## Build-up rate

The build-up rate, $R_{BU}$, was computed for each cell and each trial by calculating the excursion in firing rate from the initial response to the peak, $R_p - R_{on}$, and dividing it by the time interval $T_p - T_{on}$ between the onset of the rise and the time of peak response. For the peak response, $R_p$ and $T_p$ are as defined above, whereas for the onset of activity, $R_{on}$ and $T_{on}$, are as follows.

The onset time corresponds to the latency of the cell's response, that is, to the time point between target onset and $T_p$ at which the activity of the neuron starts ramping up. This latency, $T_{on}$,

was computed separately for each trial using the method developed by *Rowland et al. (2007)*. The firing rate $R_{on}$ was calculated by counting the spikes in a time window centered on $T_{on}$ and dividing this by the window length (50 ms).

Several other methods were tested for computing the build-up rate in individual trials. The results reported (*Figure 4b*) were robust across methods.

## Neuronal classification and selection

The 132 FEF neurons were classified by comparing their responses (mean firing rate in windows of 100–250 ms) during fixation, during the RT interval, and after the saccade. Multiple-comparison tests were performed by ANOVA. Accordingly, cells that were maximally active before the go signal were classified as fixation neurons ($n = 12$); cells that responded significantly above baseline, but only after the saccade, were deemed postsaccadic ($n = 18$); and neurons that began to respond shortly after the go signal and that were still significantly active after the saccade were deemed wide-profile ($n = 10$). The latter group could conceivably have been included in the visuomotor category described below, but given their peculiar lack of sensitivity to the saccade, they were analyzed separately.

The rest of the neurons ($n = 92$) had standard visuomotor properties and responded significantly above baseline between the go signal and saccade onset. A visuomotor index, which was equated to the coefficient $\beta_1$ in *Equation 4*, was used to characterize them (*Figure 9g*). This coefficient naturally serves as a visuomotor index because it describes the degree to which a neuron is activated by a stimulus in its RF (in which case $\beta_1 \approx 0$) as opposed to an eye movement toward it (in which case $\beta_1 \approx 1$) — which is the classic criterion used to classify FEF cells (*Bruce and Goldberg, 1985*). So, based on their $\beta_1$ values, the remaining 92 neurons were classified as either visual (V; $n = 26$), visuomotor (VM; $n = 43$), or motor (M; $n = 23$). As expected from previous studies (*Bruce and Goldberg, 1985*; *Costello et al., 2013*), during delayed-saccade trials, the neurons thus classified as V responded briskly to the presentation of the target stimulus in the RF and gradually decreased their activity thereafter; M neurons showed no activity linked to stimulus presentation but responded intensely just before movement onset; and the cells in the VM group showed both visual and presaccadic activation.

Two of the 92 neurons with standard visuomotor properties lacked well-defined RFs and were excluded from the population averages. Of the remaining 90 units, only the 62 that had error trials (IOO, OII) were considered for comparisons between correct and incorrect responses (*Figure 3*). Further comparisons between the model and neural populations presented in the main text were based on 84 neurons, with the 6 units with the lowest visuomotor indices ($\beta_1 < 0.1$) also being excluded from the main pool of 90, as these were unlikely to carry any trace of the motor signal described by the model. However, these 6 cells were included in all single-cell analyses and when explicitly dividing the population according to visual versus motor activity (*Figure 3—figure supplement 1*).

## RT matching

To tease apart the effects of RT and spatial bias on FEF activity, we devised a procedure for equalizing the RT distributions of the congruent and incongruent conditions. For each recorded cell, the observed distributions in III and IOI trials (*Figure 6a*) were sub-sampled as follows. An III trial was selected randomly and the IOI trial with the most similar RT was identified; then, if the RT difference was smaller than 15 ms, the two trials were accepted into the respective sub-samples and removed from the original pools, or else the III trial was discarded. After probing all of the III trials like this, the resulting sub-sampled pools (*Figure 6b*) contained equal numbers of trials with virtually identical RT sets. To account for variations due to random resampling, all results based on RT matching were repeated 50 times and averaged.

## Saccadic competition model

The model consists of two populations of FEF neurons that trigger saccades toward locations $T$ (where the target stimulus appears) and $D$ (diametrically opposite to $T$), their activities represented by variables $R_T$ and $R_D$. After the go signal is given, both motor plans begin to increase, and the first one to reach a threshold $\Theta$ wins the competition, thus determining the direction of the evoked

saccade and the RT. If $R_T$ wins, the saccade is correct, toward $T$, whereas if $R_D$ wins, the saccade is incorrect, toward $D$. Each simulated race corresponds to one trial of the 1DR task. The actual trajectories followed by $R_T$ and $R_D$ in each trial are dictated by the dynamics and interactions described below. MATLAB scripts for running the model are provided as supplementary source code files.

In each simulated trial, three key quantities need to be specified before the race between $R_T$ and $R_D$ can take place: the baseline firing levels, which serve as the initial values for $R_T$ and $R_D$, the initial build-up rates of the two motor plans, and the threshold, $\Theta$. Simulated neural responses are scaled so that the firing activity at threshold is around 1. The baselines for the target and distracter locations, $B_T$ and $B_D$, are specified first, drawn according to the following expressions,

$$B_T = \langle B_T \rangle \left[1 + \sigma\, \epsilon_T\right]_0$$

$$B_D = \langle B_D \rangle \left[1 + \sigma\, \epsilon_D\right]_0 \tag{5}$$

where the variability across trials is determined by $\sigma = 0.28$, and $\epsilon_T$ and $\epsilon_D$ are random Gaussian samples (different for each trial) with negative correlation equal to $-0.5$, zero mean, and unit variance. Here and in the expressions below, the square brackets with a subscript indicate that there is a minimum floor value beyond which the argument cannot drop, that is, $[x]_a = \max\{x, a\}$. This ensures, for instance, that quantities commensurate with firing activity are not negative. The mean baseline levels in **Equation 5**, $\langle B_T \rangle$ and $\langle B_D \rangle$, are set to be consistent with the spatial bias: in incongruent trials $\langle B_T \rangle = 0.16$ and $\langle B_D \rangle = 0.34$, so the target side has the lower baseline on average, whereas in congruent trials $\langle B_T \rangle = 0.34$ and $\langle B_D \rangle = 0.16$, so the target side takes the higher value. Stated differently, the higher mean baseline is always assigned to the rewarded location.

Once the baselines are drawn, the other key quantities, the threshold and the initial build-up rates, can be set for the trial. The threshold is given by

$$\Theta = \left[1.185 + 1.2\,(B_T - B_D)\right]_{0.73} \tag{6}$$

where $\Theta$ cannot drop below 0.73, as indicated by the square brackets. This expression means that the threshold for triggering a saccade increases with the baseline level on the target side and decreases with the baseline level on the opposite side, but cannot be less than a certain minimum.

The initial build-up rates (or gains) of the motor plans also depend on the baselines. For the internally driven plan, $R_D$, the build-up rate is

$$G_D = 0.001\left[1.4 + 1.7\,(B_D - B_T)\right]_0 \tag{7}$$

so the activity favoring $D$ rises most steeply when the baseline at that location is high and the baseline at the target location is low. For the target-driven motor plan, $R_T$, the rise in activity has two distinct regimes. In the first one, when $B_T \geq B_D$, the initial build-up rate is given by

$$G_T = 0.001\,(6.16 + 0.55\,\eta + 2.5\,B_T) \tag{8}$$

where $\eta$ is a random Gaussian sample (zero mean, unit variance) that varies stochastically across trials and represents intrinsic, baseline-independent fluctuations in build-up rate. In the second regime, when $B_T < B_D$,

$$G_T = 0.001\,\frac{3.0 + 0.3\,\eta + 23.25\,B_T}{1 + 1.3\,B_D} \tag{9}$$

where the numerator has the same form as in **Equation 8** but now $B_D$ appears in the denominator. The rationale for using two distinct expressions for $G_T$ is simply that the rise of the target-driven activity is very different when $R_T$ is already above the competition before the race begins compared to when it starts below the competition (**Figure 3a,b**). In the former case ($B_T \geq B_D$, regime 1) the rise is always steep, the dependence on $B_T$ is weak, and there is no further opposition from the $D$ plan (note the absence of a $B_D$ term in **Equation 8**). By contrast, when the target side starts at a disadvantage ($B_T < B_D$, regime 2), the initial build-up rate depends strongly on the actual baseline level, $B_T$, and there is sizable competition from the $D$ plan, instantiated as divisive suppression (note the dependence on $B_D$ in the denominator of **Equation 9**).

It is important to realize that, because the baselines fluctuate across trials (*Equation 5*), in general, *Equation 8* applies most often, but not uniquely, to congruent trials. Similarly, *Equation 9* applies most often, but not uniquely, to incongruent trials. The build-up rate $G_T$ depends only on the baseline values themselves, regardless of the label assigned to the spatial configuration of each trial. In other words, the local competition process has no knowledge of what determines the baselines, it simply takes them as input and evolves accordingly.

The main variables, $R_T$ and $R_D$, which represent the activities of the competing populations, are updated in each time step $\Delta t$ (set to 1 ms) as follows

$$R_T(t+\Delta t) = R_T(t) + \Delta t\, V_T$$

$$R_D(t+\Delta t) = R_D(t) + \Delta t\, V_D \tag{10}$$

where $V_T$ and $V_D$ are the instantaneous build-up rates. Initially (that is, during fixation), the activities are equal to their respective baseline values, and they remain constant until the target/go signal is presented (at $t=0$). Thus, the initial conditions are $R_T = B_T$, $R_D = B_D$, $V_T = 0$ and $V_D = 0$. The motor plans begin to advance thereafter, but not right away, because there is an afferent delay between the go signal and the actual onset of ramping activity. The target-driven plan, $R_T$, begins to rise after a short delay $A_T = 35$ ms, and does so with the build-up rate prescribed by *Equation 8* or 9, whichever applies (which can be coded as: if $t \geq A_T$, then $V_T = G_T$). The bias-driven plan, $R_D$, begins to rise after a slightly longer delay $A_D = 50$ ms, but at the beginning, it is partially inhibited by the cue presentation. During this partial inhibition, which occurs between $I_{\mathrm{on}} = 40$ and $I_{\mathrm{off}} = 155$ ms, $R_D$ rises slowly, at 38% of its nominal build-up rate, $G_D$ (if $t \geq A_D$ and $t \in [I_{\mathrm{on}}, I_{\mathrm{off}}]$, then $V_D = 0.38\,G_D$). This dynamic is based on evidence indicating that the abrupt appearance of a visual stimulus, the target in this case, briefly interrupts or suppresses ongoing saccade plans (reviewed by *Salinas and Stanford, 2018*), a phenomenon known as 'saccadic inhibition' or the 'remote distracter effect'. After the inhibition period has elapsed, the bias-driven activity may rise in full force (if $t > I_{\mathrm{off}}$, then $V_D = G_D$).

The two motor plans then continue to advance until one of them reaches threshold. However, their build-up rates may change mid-flight as one plan goes past the other. These changes are dictated by two rules that describe the two possible ways in which the competition may end.

Rule 1 (T wins): if the target-driven firing rate, $R_T$, exceeds the competing one at any point after its afferent delay has elapsed, then two things happen. First, $R_D$ is fully suppressed, so it stops increasing altogether (if $t > A_T$ and $R_T > R_D$, then $V_D = 0$). And second, the build-up rate of the $T$ plan is adjusted so that

$$V_T^{\mathrm{win}} = -0.0088 + 2.6\,G_T \tag{11}$$

(if $t > A_T$ and $R_T > R_D$, then $V_T = V_T^{\mathrm{win}}$). In this case, $R_T$ wins the race and the evoked saccade is correct, toward the target. The coefficients in *Equation 11* are such that $V_T^{\mathrm{win}}$ is typically smaller than $G_T$. This means that the target-driven motor plan typically slows down after it overtakes the competition, and the lower its initial build-up rate, the more it slows down. This change in build-up rate represents the difficulty, or cost, of resolving the conflict for the $T$ plan.

Rule 2 (D wins): if the internally driven firing rate, $R_D$, exceeds the competing one at any point after its afferent delay has elapsed and outside of the transient inhibition interval, then $R_T$ can no longer advance past $R_D$. In this case, $R_D$ simply continues to rise, winning the race without any further change in its build-up rate. The evoked saccade is incorrect, away from the target. The target-driven motor plan also keeps rising, but may suffer a minimal amount of suppression (the amount needed to ensure that $R_T$ stays below $R_D$ for the remainder of the trial).

Finally, to account for the characteristic fall in activity seen postsaccadically in FEF, after the winner reaches threshold, both motor plans decay exponentially toward a firing level of 0.2 with a time constant of 120 ms.

Note that the model generates different outcomes and RTs based on just three quantities (random numbers) that vary across trials: $\epsilon_T$ and $\epsilon_D$, which determine the baseline firing levels (*Equations 5*), and $\eta$, which adds independent noise to the build-up rate of $R_T$ (*Equations 8, 9*). There are no other sources of variability.

## Correspondence between data and model parameters

All parameter values in the model were adjusted to fit the experimental data. Each parameter typically has multiple effects, often on both the behavioral and neurophysiological responses simulated. For example, the afferent delays determine the short pause between the go signal and the onset of the rise to threshold (seen experimentally in *Figure 3a–c*, left panels), but they also pin the left tails of the RT distributions (*Figure 5g–i*) because they determine the shortest possible RTs. Similarly, the variance of the baselines ($\sigma$) generally determines the correlation between activity and RT (*Figures 3* and *7*), but it also influences the widths of the RT distributions during incongruent trials (*Figure 5h, i*).

With this in mind, note that the parameters in *Equations 5, 6* were primarily set to match the baseline and threshold values measured from the FEF population across conditions (*Figure 5—figure supplement 1*). The parameters in *Equations 7, 8, and 9* mainly determine the shapes of the RT distributions for incorrect incongruent, correct congruent, and correct incongruent trials, respectively (*Figure 5g–i*). The parameters that describe the stimulus-driven suppression of $R_D$ determine the frequency of incorrect saccades, the shape of the corresponding RT distribution, and the shape of the simulated response trajectories in those incorrect trials. Finally, the parameters associated with Rule 1 scale the RT distribution for correct incongruent trials, and also determine the steepness of the simulated target-driven rise in activity.

In all, 22 model parameters were adjusted to satisfy 23 basic experimental constraints: six baseline and six threshold values across conditions (*Figure 5—figure supplement 1*), two error rates, and three RT distributions (*Figure 5g–i*), each of which requires, at a minimum, three parameters to be characterized. But note that the model accounted for many more features (i.e., degrees of freedom) in the data, pertaining to the specific shapes of response trajectories, the effect of RT equalization, the correlation between firing activity and RT before and after target onset, and the response heterogeneity across individual FEF neurons.

---

# Additional information

### Competing interests

Emilio Salinas: Reviewing editor, *eLife*. The other authors declare that no competing interests exist.

### Funding

| Funder | Grant reference number | Author |
|---|---|---|
| National Eye Institute | R01EY12389 | Terrence R Stanford |
| National Science Foundation | Graduate Research Fellowship | Christopher K Hauser |
| National Institute of Neurological Disorders and Stroke | Training grant T32NS073553-01 | Christopher K Hauser |
| National Institute on Drug Abuse | R01DA030750 | Terrence R Stanford Emilio Salinas |
| National Eye Institute | R01EY12389-S1 | Terrence R Stanford |
| National Eye Institute | R01EY021228 | Terrence R Stanford Emilio Salinas |

The funders had no role in study design, data collection and interpretation, or the decision to submit the work for publication.

### Author contributions

Christopher K Hauser, Software, Formal analysis, Validation, Investigation, Visualization, Methodology, Writing—original draft, Writing—review and editing; Dantong Zhu, Investigation, Data acquisition; Terrence R Stanford, Conceptualization, Resources, Data curation, Supervision, Funding acquisition, Investigation, Methodology, Writing—original draft, Project administration, Writing—review and editing; Emilio Salinas, Conceptualization, Resources, Data curation, Software, Formal

analysis, Supervision, Funding acquisition, Validation, Investigation, Visualization, Methodology, Writing—original draft, Project administration, Writing—review and editing

### Author ORCIDs

Emilio Salinas (iD) http://orcid.org/0000-0001-7411-5693

### Ethics

Animal experimentation: Experimental subjects were two adult male rhesus monkeys (Macaca mulatta). All experimental procedures were conducted in accordance with the NIH Guide for the Care and Use of Laboratory Animals, USDA regulations, and the policies set forth by the Institutional Animal Care and Use Committee (IACUC) of Wake Forest School of Medicine under protocols A10-192, A13-088 and A16-192. All surgery was performed under sodium pentobarbital anesthesia, and every effort was made to minimize suffering.

### Decision letter and Author response

Decision letter https://doi.org/10.7554/eLife.33456.027
Author response https://doi.org/10.7554/eLife.33456.028

## Additional files

### Supplementary files

• Source code 1. MATLAB script for simulating the simplest version of the saccadic competition model. This script generates RT distributions and average $R_T$ and $R_D$ traces as functions of time (as in *Figure 5*).
DOI: https://doi.org/10.7554/eLife.33456.021

• Source code 2. MATLAB script for simulating the saccadic competition model with two types of target-driven response. This script works in the same way as *Source code 1*, except that $R_T$ is split into separate $R_T^C$ and $R_T^U$ components (as defined in *Equation 1*).
DOI: https://doi.org/10.7554/eLife.33456.022

• Source code 3. MATLAB script for sorting the response traces produced by *Source code 1* and *2* into RT quantiles (as in *Figures 7* and *10*).
DOI: https://doi.org/10.7554/eLife.33456.023

• Transparent reporting form
DOI: https://doi.org/10.7554/eLife.33456.024

### Data availability

Matlab scripts for running the model are provided as supplementary code files. Experimental data are available from the corresponding author (esalinas@wakehealth.edu) upon reasonable request.

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
