## [Decision Letter]

Thank you for submitting your article "Motor selection dynamics in FEF explain the reaction time variance of saccades to single targets" for consideration by *eLife*. Your article has been favorably evaluated by Timothy Behrens (Senior Editor) and three reviewers, one of whom is a member of our Board of Reviewing Editors. The reviewers have opted to remain anonymous.

The reviewers have discussed the reviews with one another and the Reviewing Editor has drafted this decision to help you prepare a revised submission.

Summary:

In this paper, the authors explored neural activity in the frontal eye field (FEF) in relation to trial-to-trial variability in saccadic reaction times (RT). They used a simple visually guided saccade task, but with the manipulation that one of the saccade directions could have been rewarded more than the other potential directions. This manipulation creates a bias due to reward expectancy that affects RT distributions. The authors explored FEF neural activity in relation to such variability. A major component of the work describes a model that instantiates interactions between an internally represented spatial preference (in this case, where a monkey really wants to look to get a reward) and where a visual target is actually presented. The resolution of these interactions determines where the monkey eventually looks. The comparison of the profile of activation within the model match that observed with neural recordings, as do the simulated and observed profile of RTs.

The neural results show that the location with high reward expectancy has higher baseline activity before target onset than other locations, as might be expected. According to the model, it is essentially baseline activity that dictates the rest of the activity dynamics and the resultant RT distributions.

Essential revisions:

The reviewers all agree that the questions are timely and interesting. Moreover, the idea that it is the baseline activity that matters is intriguing, and so there is value in the model.

However, the reviewers also had a number of major concerns.

1) All three reviewers raised serious concerns about the readability of the manuscript. As one of the reviewers noted, "I had a very hard time understanding a lot of sections of it, and it was therefore difficult to evaluate this paper. I would really like that the authors work much more on clarity in their revision." Another noted that "the text is overly dense in places, overly long-winded in others. Throughout, the key points feel like they are buried under too many details and too few clear demonstrations of the key features of the data that support the model."

Below is a compilation of more specific suggestions provided by the reviewers concerning clarity and readability:

a) Abstract: It is hard to understand why strong co-dependencies imply that intrinsic randomness contributes little to saccade variance. It is also not entirely clear what is meant by non-linear co-dependencies. For a reader who has not read the rest of the paper, these two critical sentences of the Abstract are very difficult to understand. I would recommend making the Abstract, especially these two critical take-home messages, more readable to even someone from outside the field. It might even help to explain what the co-dependencies are in the first place.

b) First paragraph of Introduction: like for the Abstract, some of this paragraph was confusing. For example, why is the saccadic system "singularly" susceptible to this "confound", and what is the confound in the first place? Also, based on the Abstract (i.e. not having read the rest of the paper), the impression I get from the Abstract is that it is not randomness that causes variability in RT but instead a fundamentally deterministic mechanism. Then, in the first paragraph of Introduction, I see that people assume a "seemingly firm, mechanistic account" of variability (which to me is a sentence that can be interpreted as agreeing with the conclusion of the Abstract), but then you say that this is incorrect. Why the inconsistency? These points seem a bit clearer after reading the rest of the paper, but for a reader who starts with the Abstract and Introduction, the very first two paragraphs of the paper are quite hard to parse.

c) Last sentence of Introduction: again, the sentiment is interesting, but there seems to be a disconnect between what is told to the reader up to this point and what the sentence says. For example, the sentence uses the expression "internal biases", but this is not described earlier in the paragraph or the rest of the Introduction. So, the reader has to guess what you mean. e.g. is the concept of internal biases related to the large fluctuations of "baseline activity" that was alluded to earlier in the paragraph? Some more simple and clear description throughout the Abstract and Introduction would greatly help this paper reach a wide audience. For example, something like “if saccades can be very fast, why, under identical stimulation conditions, are they sometimes very slow?” should come earlier in the paper. Around the third paragraph of the subsection “Modeling the rise-to-threshold dynamics” is exactly where the take home message of the paper becomes more clear, and it would be nice if this was made much earlier on.

d) The key, novel finding, as captured by the model, is that the build-up and end-point of the motor plan depend critically on the baseline value, regardless of the task condition that caused that baseline value. It would be useful to show this empirically as clearly as possible. Using, for example, linear regression, how are those quantities related per-neuron and per-condition?

e) It would then be useful to show that the key dynamics of the model (especially Equations 6-9) come directly from these empirical relationships or more generally explain where all the constants in those equations come from and if/how the findings are robust to the particular values used. Also, to what extent is the coupling between baseline and both rate-of-rise and the endpoint necessary to produce the measured results?

f) Finally, explain how heterogeneity in the population of neural responses can be reconciled with the actual RTs that are produced; i.e., what range/proportions of response properties need to be in the population for it to produce the measured errors and RTs? Do the baseline values need to be coupled across neurons for the population to produce a predictable choice/RT on a given trial?

2) A number of questions were raised about the strength of support for the conclusion that baseline activity plays such a primary role in driving the saccade-generation process. Specifically:

a) One of the key observations in this manuscript is of co-dependencies between various parameters. However, there seems to be a (perhaps unintended) primacy given to the baseline rate, seen for example in statements implying that baseline activity is "directly causal" to or predictive of subsequent movement-related dynamics. The notion of causality is evocative when stated this way, inferring that everything proceeds from the starting baselines. An alternative explanation is that all movement-related parameters are influenced by some common factor like reward expectancy or spatial attention. Likewise, reward expectancy is likely not a binary concept, but would itself depend on the animal's level of satiation during a session. Incorporating such internal (and somewhat unknowable) variables complicates the "simplest non-trivial problem" to a considerable degree. In general, it is not clear if the conclusions drawn from the results apply only to a task that is by intent designed to strongly manipulate an internal drive (and hence relates to the question of generalizability, details below).

b) Many of the analyses in the paper rely on a degree of averaging either across the recorded sample (e.g., Figure 3), or within neurons. There are some trial-by-trial analyses that relate baseline to RT, but I did wonder why trial-by-trial correlations between baseline and other saccade-related parameters such as build-up rate and threshold were not explicitly presented. Claims are made (e.g., subsection “Modeling the rise-to-threshold dynamics”, third paragraph) that the variance in buildup rate and threshold stems from baseline at a trial-by-trial level, but I do not believe this was shown at this degree of resolution. Related to this, please separate any trial-by-trial analyses to the functional class of a neuron (e.g., for V, VM, and M-cells); my reasoning here is that baseline seems to barely change in the M-dominant cells shown in Figure 3—figure supplement 1.

c) In all of the figures with fine RT bins, it is ultimately the "visual response" that appears to be most correlated with RT and baseline fluctuations (e.g. Figure 6, 8, and so on). That is, short RT trials have strong early visual bursts, and long RT trials have weak and late visual bursts. The rest just follows. So, maybe the baseline activity dictates the quality of the visual response, and the rest just follows? The authors do not mention any of this in their paper. Of course, with visually guided saccades having very short RT, the visual and motor bursts are hard to distinguish; however, their reward manipulation was done exactly to increase RT, and it is very clear that in a lot of their neural and model results, clear effects on visual bursts alone can be recovered. Can the authors therefore comment on the importance of the visual burst on RT dynamics in their model and data? This idea would actually be consistent with brainstem ideas (e.g. Boehnke and Munoz, 2008, On the importance of visual transients. Also, Chen and Hafed, JNP, 2017 showed that the visual burst in the superior colliculus even during fixation without any saccades can predict RT fluctuation dynamics from completely separate sessions involving saccades). It would also be in line with some ideas from smooth pursuit eye movements (e.g. Lisberger) in which it is sensory variability that almost entirely dictates final motor variability.

3) How generalizable are the results? As the authors cite in this manuscript, there have been any number of studies that have related neurophysiological recordings in the FEF or SC to rise-to-threshold models and/or RT distributions. Indeed, and as found here, many of these neurophysiological results have challenged the idea of a fixed threshold, and instead related changes in RT to variations in baseline, variations in rate of rise, or variations in the onset of saccade-related accumulation (see below); all of these previous studies have relied on different behavioral tasks.

Thus, a key question is: Is the framework proposed here specific to a task that manipulates spatial reward expectancy, uses a high degree of temporal predictability (e.g., the fixation period is fixed at 1,000 ms), and/or requires selection between an internal bias and an externally-cued alternative? The authors may be able to resolve this to some degree by analyzing data collected from the "all direction rewarded" condition, or perhaps even from the delayed saccade task. These tasks can also feature substantial RT variability; would they also show co-dependencies in baseline activity, buildup rate, and threshold? If possible, it would be very informative to see FEF data from these other tasks analyzed like the data shown in Figures 6, 8, and 9.

Other works that have investigated this issue have stressed the importance of the time of onset of saccade-related accumulation in determining RT variance (e.g., Pouget, Loga et al., 2011 in the context of adaptive RT adjustments; Peel et al., 2017 in the context of delayed response tasks; see also the review by Teichert, Grinwald and Ferrera, J Neurophys 2016). In fact, in the Pouget and Peel papers, changes in onset of accumulation, rather than changes in baseline, rate of rise, or threshold, best explained associated changes in RT, even down to the trial-by-trial level; the Pouget paper seems particularly relevant, given that the behavior in that paper (featuring a reactive saccade generated after an error) also involves resolution of an internal bias (in this case to slow down) with an externally-cued alternative. Given this, I was surprised that onset was not considered in the current paper. Indeed, it seems like the onset of activity is varying in the neural activity shown in Figure 7A (3rd row), and in Figure 8A and 8D. In reactive tasks, the onset of accumulation may be particularly important (or easier to extract) for predominantly movement-related neurons (see also the work of Purcell, Schall and colleagues). Some discussion, and ideally analysis, of onset of saccade-related activity would seem to be warranted.

An assumption throughout the manuscript is that the saccades made in the various conditions are the same from the perspectives of saccade metrics or kinematics; at the end of the subsection “Behavioral manifestations of a spatial bias” it is stated explicitly that a conflict between target and reward location will play out in saccade direction and timing. My sense is that this is unlikely to be the case; previous work with the 1-DR task emphasized that saccades to rewarded locations are also faster than those to unrewarded locations (see for example Takikawa, Kawagoe, Itoh, Nakahara and Hikosaka, Exp Brain Res 2002); saccade metrics and kinematics can also influence the firing of SC neurons (e.g., Ikeda and Hikosaka, J Neurophys 2007). My suggestion is that the authors describe any changes in metrics (e.g., endpoint scatter) and velocity for saccades to rewarded vs. unrewarded locations, and consider how any dependencies may or may not influence the interpretation of their data. To be clear, it may be that the FEF is simply far enough upstream that such considerations are not pertinent, but I would like to hear the authors' views on this.

Finally, how do the results relate to recent results from the laboratory of Brian Corneil reversibly inactivating the FEF and exploring SC activity dynamics and RT variability?

---

## [Author Response]

Essential revisions:1) All three reviewers raised serious concerns about the readability of the manuscript. As one of the reviewers noted, "I had a very hard time understanding a lot of sections of it, and it was therefore difficult to evaluate this paper. I would really like that the authors work much more on clarity in their revision." Another noted that "the text is overly dense in places, overly long-winded in others. Throughout, the key points feel like they are buried under too many details and too few clear demonstrations of the key features of the data that support the model."

We really appreciate all the specific suggestions. Also, the manuscript has been revised at various points with this comment in mind.

Below is a compilation of more specific suggestions provided by the reviewers concerning clarity and readability:a) Abstract: It is hard to understand why strong co-dependencies imply that intrinsic randomness contributes little to saccade variance. It is also not entirely clear what is meant by non-linear co-dependencies. For a reader who has not read the rest of the paper, these two critical sentences of the Abstract are very difficult to understand. I would recommend making the Abstract, especially these two critical take-home messages, more readable to even someone from outside the field. It might even help to explain what the co-dependencies are in the first place.

Got it. We have rewritten the second half of the Abstract to better integrate the take-home messages:

“In studies of voluntary movement, a most elemental quantity is the reaction time (RT) between the onset of a visual stimulus and a saccade toward it. […] The ensuing conflict resolution process, which manifests via subtle covariations between baseline activity, build-up rate, and threshold, consists of fundamentally deterministic interactions, and explains the observed RT distributions while invoking only a small amount of intrinsic randomness.”

We think this now presents a single, more precise and more nuanced conclusion.

b) First paragraph of Introduction: like for the Abstract, some of this paragraph was confusing. For example, why is the saccadic system "singularly" susceptible to this "confound", and what is the confound in the first place? Also, based on the Abstract (i.e. not having read the rest of the paper), the impression I get from the Abstract is that it is not randomness that causes variability in RT but instead a fundamentally deterministic mechanism. Then, in the first paragraph of Introduction, I see that people assume a "seemingly firm, mechanistic account" of variability (which to me is a sentence that can be interpreted as agreeing with the conclusion of the Abstract), but then you say that this is incorrect. Why the inconsistency? These points seem a bit clearer after reading the rest of the paper, but for a reader who starts with the Abstract and Introduction, the very first two paragraphs of the paper are quite hard to parse.

We agree; thanks for going carefully over this. The revised version now says:

“In the case of eye movements, this ambiguity is likely more severe than previously appreciated. There is a firm, mechanistic account describing how saccades are triggered, but according to the present results, that account lacks a crucial ingredient – ongoing motor conflict – and assumes, incorrectly, that in response to the same stimulus, the fundamental reason why some saccades are triggered very quickly whereas others take much longer simply boils down to noise in the underlying neuronal activity.”

This now pinpoints exactly what is missing from the established account, and reinforces the conclusion stated in the Abstract. Also, it incorporates some of the language suggested in the next point.

c) Last sentence of Introduction: again, the sentiment is interesting, but there seems to be a disconnect between what is told to the reader up to this point and what the sentence says. For example, the sentence uses the expression "internal biases", but this is not described earlier in the paragraph or the rest of the Introduction. So, the reader has to guess what you mean. e.g. is the concept of internal biases related to the large fluctuations of "baseline activity" that was alluded to earlier in the paragraph? Some more simple and clear description throughout the Abstract and Introduction would greatly help this paper reach a wide audience. For example, something like “if saccades can be very fast, why, under identical stimulation conditions, are they sometimes very slow?” should come earlier in the paper. Around the third paragraph of the subsection “Modeling the rise-to-threshold dynamics” is exactly where the take home message of the paper becomes more clear, and it would be nice if this was made much earlier on.

Thanks for identifying the specific issues to address. The last paragraph of the Introduction now ends as follows:

“A model based on competitive dynamics quantitatively reproduced the temporal profiles of the evoked neural responses, as well as their dependencies on RT, reward expectation, and trial outcome (correct/incorrect) – this, while simultaneously matching the monkeys’ full RT distributions across experimental conditions. […] Thus, the noise in the build-up rate is much more modest than predicted by extant frameworks, and the high variability of saccades to single targets is, to a large degree, deterministic, a direct consequence of motor selection mechanisms that allow voluntary saccades to be driven by both sensory events and internal biases.”

This is very much in the spirit of the take home message identified by the reviewer, explains what we mean by “internal biases,” and is consistent with the phrasing of the Abstract.

d) The key, novel finding, as captured by the model, is that the build-up and end-point of the motor plan depend critically on the baseline value, regardless of the task condition that caused that baseline value. It would be useful to show this empirically as clearly as possible. Using, for example, linear regression, how are those quantities related per-neuron and per-condition?

Great idea. We had initially carried out extensive linear regression analyses, but prior to the model the relationships between the neural responses, RT, and reward condition were difficult to disentangle and interpret. We have included a new figure, now Figure 4, that demonstrates two things, a very strong association between build-up rate and RT, and a strong association between baseline activity and peak response at the level of single neurons. These data make it easier to set up the model. The ninth paragraph of the subsection “Motor conflict during the rise-to-threshold process” describes these data.

e) It would then be useful to show that the key dynamics of the model (especially Equations 6-9) come directly from these empirical relationships… or more generally explain where all the constants in those equations come from and if/how the findings are robust to the particular values used.

The data in the new Figure 4 demonstrate key qualitative relationships, but nothing more specific that can be directly taken as the value of a model parameter, unfortunately (for several reasons; see next response). The constants in those equations (and Equation 11) were adjusted so that the model would fit both the behavioral data and the neural data in Figure 5. This is explained in the last section of the Materials and methods, and is now explicitly mentioned in the main text as well (subsection “Modeling the rise-to-threshold dynamics”, eighth paragraph).

Regarding the issue of robustness, yes, the results vary in a graded way with all the parameters (and even, to some extent, with the form of the equations). This is now addressed in a new short subsection titled “Model robustness”, and in in the new Figure 7—figure supplement 2.

Also, to what extent is the coupling between baseline and both rate-of-rise and the endpoint necessary to produce the measured results?

Absolutely critical. The empirical relationships demonstrate correlations that are relatively weak (new Figure 4), and yet, according to the model, underlie strong deterministic dependencies. How is that possible? The *observed* correlations are weak because (1) single neurons are noisy, (2) baseline firing levels are very low, which amplifies the noise problem, (3) there is considerable heterogeneity in the actual neural population, and (4) they do not take into consideration the effect of the competing motor plans. This is exactly why the model is so important.

f) Finally, explain how heterogeneity in the population of neural responses can be reconciled with the actual RTs that are produced; i.e., what range/proportions of response properties need to be in the population for it to produce the measured errors and RTs?

This is an interesting question, as those proportions (of fast- and slow-preferring neurons) could potentially provide another constraint for the model. However, the model is agnostic about this. The choices and RTs are determined by the summed activity of the two components, *R_T_^C^*and *R_T_^U^*, each of which, as suggested by the reviewer, could indeed be thought of as a sum over individual neurons — but these would be *weighted* sums. If we assumed that there were, say, twice as many *C* neurons as *U* neurons, we could counter that by also assuming that their synaptic weights were half as strong as those of the *U* neurons, so that the factor *R_T_^C^*would not change. In other words, it is impossible to say something about the numbers of neurons of each type without making additional assumptions about their relative impact.

Do the baseline values need to be coupled across neurons for the population to produce a predictable choice/RT on a given trial?

The requirement is very loose. In the model, the *C* component of the target-driven response absorbs the main dependency on the baseline (Equation 3). Again, thinking of *R_T_^C^*as a weighted sum over neurons, each *C* neuron could have a different mean baseline, and their fluctuations could be independent to varying degrees, as long as the sum over cells produced fluctuations in *R_T_^C^*of the appropriate mean and variance. There is a lot of leeway in this case, especially if one is again willing to consider a distribution of weights across the population.

Both of these questions hint at the idea of expanding the model so that it consists of an actual population (or two) of *diverse*, single neurons. There is certainly room for that, and is something that we will be working on, because some interesting aspects of our data could be matched to that type of model. But that is beyond the scope of the present manuscript, where only two qualitatively different types of response are distinguished.

2) A number of questions were raised about the strength of support for the conclusion that baseline activity plays such a primary role in driving the saccade-generation process. Specifically:a) One of the key observations in this manuscript is of co-dependencies between various parameters. However, there seems to be a (perhaps unintended) primacy given to the baseline rate, seen for example in statements implying that baseline activity is "directly causal" to or predictive of subsequent movement-related dynamics. The notion of causality is evocative when stated this way, inferring that everything proceeds from the starting baselines. An alternative explanation is that all movement-related parameters are influenced by some common factor like reward expectancy or spatial attention. Likewise, reward expectancy is likely not a binary concept, but would itself depend on the animal's level of satiation during a session. Incorporating such internal (and somewhat unknowable) variables complicates the "simplest non-trivial problem" to a considerable degree. In general, it is not clear if the conclusions drawn from the results apply only to a task that is by intent designed to strongly manipulate an internal drive (and hence relates to the question of generalizability, details below).

We fundamentally agree with this observation. Our model shows that the baselines *could* be causal to the subsequent movement-related dynamics, but we surely cannot rule out some other common factor(s). The reference to causality spotted by the reviewer has been softened as follows:

“[…] the fluctuations in baseline at the two relevant locations are directly linked and possibly causal to the subsequent movement-related dynamics and, ultimately, to the RTs generated.” In addition, we now clarify what it is that suggests a causal relationship:

“[…] the activity in FEF demonstrated characteristic covariations in the three key features of the rising saccade-related activity. […] This suggests a causal relationship between the baseline and the subsequent response, because the baseline signal arises earlier, before target presentation, and because it is predictive of outcome (Figure 3F).”

The issue is also mentioned in the Discussion, where we explicitly state the caveat:

“This is a simplification, of course. First, it is possible that the link is not strictly causal, i.e., that some unidentified factor drives both the fluctuations in baseline and in the other dynamic variables.”

As to the comment that “reward expectancy is likely not a binary concept,” we totally agree, but also think that the tie to reward expectation is not obligatory. The reward manipulation generates a condition in which the monkey is strongly incentivized to plan an eye movement to a known location. Cueing paradigms such as those used in attention studies likely produce similar effects (consistent with reported changes in baseline activity during the deployment of spatial attention). Studies using subthreshold microstimulation (mentioned in the Discussion) are also in agreement with this. The covert planning is what is key, not the specific factor that happens to drive it. The model describes what happens when there is a conflict between a stimulus-driven and a bias-driven motor plan, whatever the source of the bias. We suggest that the corresponding oculomotor dynamics are likely to be ubiquitous precisely because under natural viewing conditions there are multiple stimuli and multiple endogenous factors (current goals, motivation, etc.) acting simultaneously and associated with numerous movement vectors, thus creating ongoing motor conflict preceding every saccade.

The issue of generalization is addressed in the subsection “Model robustness”, as is explained in more detailed in point 3, below.

b) Many of the analyses in the paper rely on a degree of averaging either across the recorded sample (e.g., Figure 3), or within neurons. There are some trial-by-trial analyses that relate baseline to RT, but I did wonder why trial-by-trial correlations between baseline and other saccade-related parameters such as build-up rate and threshold were not explicitly presented. Claims are made (e.g., subsection “Modeling the rise-to-threshold dynamics”, third paragraph) that the variance in buildup rate and threshold stems from baseline at a trial-by-trial level, but I do not believe this was shown at this degree of resolution.

Yes, those data are now shown in the new Figure 4. Please see the response to point 1d above.

Related to this, please separate any trial-by-trial analyses to the functional class of a neuron (e.g., for V, VM, and M-cells); my reasoning here is that baseline seems to barely change in the M-dominant cells shown in Figure 3—figure supplement 1.

This is an important point, for sure. The average traces mentioned by the reviewer show that the V neurons have a higher baseline level. And indeed, the magnitude of the baseline showed a slight declining trend as a function of visuomotor index (not shown). But crucially, the correlations between baseline and responsivity (i.e., peak response) shown in the new Figure 4 were equally strong across the visuomotor range . The *modulations* in baseline showed no evidence of a bias in the contribution of V neurons. See also the response to the next point.

c) In all of the figures with fine RT bins, it is ultimately the "visual response" that appears to be most correlated with RT and baseline fluctuations (e.g. Figure 6, 8, and so on). That is, short RT trials have strong early visual bursts, and long RT trials have weak and late visual bursts. The rest just follows.

We are not quite sure what is the proposed scenario here. When comparing “early” and “late,” we think the reviewer is referring to the time of the visual burst relative to saccade onset. If so, it is true that the visual burst tends to produce a correlation between firing rate and RT as the saccade onset becomes more distant from target onset. However, this effect does not explain the RT dependencies of the VM and M neurons (such as those shown in Figure 9, for which the visual burst is minimal). The temporal alignment effect of the visual bursts is *consistent* with the overall trend in IOI trials, which is for presaccadic responses to be stronger at short RTs, but it is neither necessary nor sufficient for explaining the stratification of the response trajectories with RT, as demonstrated when the V neurons are excluded from the population averages; in that case (see new Figure 7—figure supplement 2), the results are very similar to those obtained with the full population (Figure 7).

The contribution of the visual burst is a bit tricky to interpret, but it does not uniquely account for the results.

So, maybe the baseline activity dictates the quality of the visual response, and the rest just follows?

The correlation between baseline activity and responsivity seems to be pretty much the same for all cell types (Figure 4, second row; Figure 6—figure supplement 1). So, while it may be true that the baseline modulates the gain of the visual responses, the effect is by no means exclusive to the V neurons. It is not the case that the rest just follows.

The authors do not mention any of this in their paper. Of course, with visually guided saccades having very short RT, the visual and motor bursts are hard to distinguish; however, their reward manipulation was done exactly to increase RT, and it is very clear that in a lot of their neural and model results, clear effects on visual bursts alone can be recovered. Can the authors therefore comment on the importance of the visual burst on RT dynamics in their model and data?

It is important to note that the model simulates the total motor activity in FEF, a response that always starts ramping linearly with the same latency and has no separate visual burst. The actual aggregate FEF activity, which combines neurons with a diversity of temporal profiles, behaves very much like this (Figure 7). The visual bursts certainly contribute to this aggregate, but are not the exclusive (or dominant) source of the observed modulations across RTs.

This is now specifically addressed in the first paragraph of the subsection “Model robustness”, where the issue is framed in terms of the following question: to what degree do the results depend on the precise mix of V, VM, and M neurons included in the analyses? When the most visually dominant neurons are excluded (about 30% of the cells; Figures 6—figure supplement 1, Figure 7—figure supplement 2), the observed responses show almost identical modulations with RT as the full population (Figure 7). So, the magnitude of the visual component is not critical.

This idea would actually be consistent with brainstem ideas (e.g. Boehnke and Munoz, 2008, On the importance of visual transients. Also, Chen and Hafed, JNP, 2017 showed that the visual burst in the superior colliculus even during fixation without any saccades can predict RT fluctuation dynamics from completely separate sessions involving saccades). It would also be in line with some ideas from smooth pursuit eye movements (e.g. Lisberger) in which it is sensory variability that almost entirely dictates final motor variability.

Again, the data are not inconsistent with variations in the gain of the visual bursts, as documented in those papers, but that is not the main source of the RT effects in this case; here, RTs vary fundamentally because of motor competition, which is now emphasized in the Abstract and Introduction (see point 1 above).

3) How generalizable are the results? As the authors cite in this manuscript, there have been any number of studies that have related neurophysiological recordings in the FEF or SC to rise-to-threshold models and/or RT distributions. Indeed, and as found here, many of these neurophysiological results have challenged the idea of a fixed threshold, and instead related changes in RT to variations in baseline, variations in rate of rise, or variations in the onset of saccade-related accumulation (see below); all of these previous studies have relied on different behavioral tasks.

We agree that different sources of RT variance may contribute to different degrees depending on the specifics of the task at hand. In general, however, we would argue that the magnitude of the effects in our data is much larger than in previous studies, indicating that motor conflict may generate much larger variations in RT than other factors (as detailed below). Surely, when motor conflict is minimized, as in our congruent condition, those other factors may account for a sizable proportion of the remaining variance, in agreement with our own modeling results.

Thus, a key question is: Is the framework proposed here specific to a task that manipulates spatial reward expectancy, uses a high degree of temporal predictability (e.g., the fixation period is fixed at 1,000 ms), and/or requires selection between an internal bias and an externally-cued alternative? The authors may be able to resolve this to some degree by analyzing data collected from the "all direction rewarded" condition, or perhaps even from the delayed saccade task. These tasks can also feature substantial RT variability; would they also show co-dependencies in baseline activity, buildup rate, and threshold? If possible, it would be very informative to see FEF data from these other tasks analyzed like the data shown in Figures 6, 8, and 9.

This is an excellent point; thanks for bringing it up. We had thought about presenting the ADR data but were not sure about it. Framed as a question of generalization, though, it is clearly important to know whether the model can deal with the simpler, unbiased condition.

The results for the ADR blocks are shown in Figure 7—figure supplement 3 and are discussed in the last paragraph of the subsection “Model robustness”. Indeed, the model replicates the RT distribution and the neural data in the ADR task with what we consider to be the minimal parameter changes to be expected. After adjusting its mean baseline values to the measured one, all we had to do to match the data was lower the variability of the baselines (*σ* in Equation 5) (the second parameter change mentioned in the text was small and eliminated a small discrepancy in the RT distribution). The ADR neural responses are a bit noisy because we had fewer cells in that condition, but the agreement is evident. The results show that all the mechanisms inferred for the 1DR congruent and incongruent conditions are entirely consistent with the simpler, symmetric case, which, according to the model, differs simply in the degree to which the bias- and stimulus-driven plans differ from each other initially.

Other works that have investigated this issue have stressed the importance of the time of onset of saccade-related accumulation in determining RT variance (e.g., Pouget et al., 2011 in the context of adaptive RT adjustments; Peel et al., 2017 in the context of delayed response tasks; see also the review by Teichert, Grinwald and Ferrera, J Neurophys 2016). In fact, in the Pouget and Peel papers, changes in onset of accumulation, rather than changes in baseline, rate of rise, or threshold, best explained associated changes in RT, even down to the trial-by-trial level; the Pouget paper seems particularly relevant, given that the behavior in that paper (featuring a reactive saccade generated after an error) also involves resolution of an internal bias (in this case to slow down) with an externally-cued alternative. Given this, I was surprised that onset was not considered in the current paper. Indeed, it seems like the onset of activity is varying in the neural activity shown in Figure 7A (3rd row), and in Figure 8A and 8D. In reactive tasks, the onset of accumulation may be particularly important (or easier to extract) for predominantly movement-related neurons (see also the work of Purcell, Schall and colleagues). Some discussion, and ideally analysis, of onset of saccade-related activity would seem to be warranted.

We did look at the time of response onset in our recorded data but found very little evidence of systematic variations in timing in the average population activity. In general, the apparent variations in response onset almost disappear once the different baselines and build-up rates are taken into account. Consider, for instance, the comparison between congruent versus incongruent trials shown in Author response image 1. The traces are the same as in Figure 6e, but limited to a short time period around the go signal, when activity starts rising. The more steeply rising trace was rescaled and shifted vertically and horizontally to maximize the overlap between the two curves, and the optimal time shift for this was only 3 ms. Other comparisons produced similar results. The largest shift was between responses in short- versus long-RT trials in the IOI condition, but it was still only 11 ms.

Just to be clear, single neurons did show a range of temporal variations in onset, but of course, for individual cells even very large effects are entirely expected. For instance, for an ideal motor neuron the response onset could very well shift in direct proportion to the RT. So, the question is really whether the population activity shifts or not, which is what the aforementioned papers evaluated.

The study by Pouget showed evidence that response onset can change systematically, but the magnitude of the effect they reported was actually quite small. On average, they found differences of approximately 15 ms between two task conditions (conditions that should, in fact, maximize those differences given the timing constraints of the stop signal task). The difference was about 20 ms in one data set but only about 5 ms in the other, which was the one with more trials (about 3 times as many as in the first set). Variations of ∼10 ms are very small in comparison with those we are trying to account for: the mean RT difference across conditions (congruent vs. incongruent) was close to 100 ms, and the SD of the RT distributions for incongruent trials was about 80 ms.

The Discussion now summarizes this issue as follows:

“And second, other sources of variability are likely to contribute [to the observed variance] too; for instance, variations in response onset (Pouget et al., 2011; Peel et al., 2017), which would correspond to fluctuations in the afferent delays of the model. […] It is certainly possible to add noise to the afferent delays or other components of the model without substantially altering the results, but what is notable is that this is not necessary.”

The paper by Peel et al. looked at a very different (non-reactive) task in which motor conflict was probably minimized. Please see the last part of this point (point 3), below, for a more detailed discussion of that result.

An assumption throughout the manuscript is that the saccades made in the various conditions are the same from the perspectives of saccade metrics or kinematics; at the end of the subsection “Behavioral manifestations of a spatial bias” it is stated explicitly that a conflict between target and reward location will play out in saccade direction and timing.

We did not mean to make such implication or assumption; our apologies for the misunderstanding. The relevant passage now reads as follows:

“In short, monkeys are highly sensitive to the spatially asymmetric value associated with otherwise identical target stimuli (for additional evidence of this, see Figure 2—figure supplement 1). This manifests primarily as large differences in RT between correct congruent, correct incongruent, and incorrect incongruent trials (for other manifestations, see Figure 2— figure supplement 2).”

We emphasize the variations in RT because they are enormous (∼three-fold changes in SD) compared to those in peak velocity (variations of ∼5%) or other metrics, but now we show the data (in the new Figure 2— figure supplement 2) and make it clear that the latter are not excluded.

My sense is that this is unlikely to be the case; previous work with the 1-DR task emphasized that saccades to rewarded locations are also faster than those to unrewarded locations (see for example Takikawa, Kawagoe, Itoh, Nakahara and Hikosaka, Exp Brain Res 2002); saccade metrics and kinematics can also influence the firing of SC neurons (e.g., Ikeda and Hikosaka, J Neurophys 2007). My suggestion is that the authors describe any changes in metrics (e.g., endpoint scatter) and velocity for saccades to rewarded vs. unrewarded locations.

Our data (shown in the new Figure 2—figure supplement 2) are entirely consistent with those previous reports, which are now referenced in the caption of the figure.

And consider how any dependencies may or may not influence the interpretation of their data. To be clear, it may be that the FEF is simply far enough upstream that such considerations are not pertinent, but I would like to hear the authors' views on this.

We think that there is no inconsistency really, and that saccade metrics can only affect the interpretation minimally. Critically, there is no fixed relationship between variations in RT and in saccade metrics, and therefore, the (neural) cause of the former can hardly be constrained by measurement of the latter. The same conclusion is reached if we look at this the other way around, that is, if we start from the assumption that the same neural activity (in FEF or SC) influences both the timing and the metrics of saccades. In that case, if we claim that certain variations in neural activity are responsible for the observed variations in RT, then we can expect that the same activity will also produce *some* variations in saccade metrics. But again, it is difficult to go much further given the loose association between RT and saccade metrics.

It is worth mentioning, however, that based on other recent work from our lab (currently under review), we have a working hypothesis: that whereas the threshold-crossing event determines saccade onset, the build-up rate of the same population activity at the time of threshold crossing is what influences the peak velocity of the impending saccade (i.e., higher build-up rate translates into higher peak velocity). This is speculative at this point, but it is a specific example of a possible link between RT and peak velocity mediated by FEF activity, and one that is consistent with the 1DR results.

Finally, how do the results relate to recent results from the laboratory of Brian Corneil reversibly inactivating the FEF and exploring SC activity dynamics and RT variability?

In the recent study from Brian Corneil’s lab (Peel et al., 2017) they found that the dominant effect of FEF inactivation was to increase saccadic RT and the onset of SC activity, rather than the build-up rate or other parameters of the ensuing motor plan. In our data, the onset of the rise to threshold varied very little across experimental conditions and RT bins, at least in comparison with the variations in other response features (see Author response image 1).

At least one crucial difference between the studies is that the inactivation results were based on memory-guided saccades, which are not reactive and are triggered well after the target location is known. Thus, in a way, the RTs in that task and in ours are rather different phenomena. In the task used by Peel et al., there is ample opportunity for any ongoing motor conflict to be fully resolved during the delay. So, by the time the go signal is given there is no reason for ambiguity in the choice; at that time, the motor plan toward the target can be substantially advanced already. The corresponding RT then simply reflects the speed with which the trigger can be pulled, so to speak, a process that may be generally slower in that case because the triggering event (offset of fixation point) is not spatially overlapping with the onset of the saccade target. Under those conditions, it makes sense that detection of the visual offset at fixation (which partly determines the onset of the rising activity) would dominate the variability of the measured RTs, as other sources of variance associated with motor planning would likely be minimized. In contrast, in our task a strong spatial conflict arises at the time of the go signal, and the observed RT largely reflects the amount of time needed for it to be resolved. In this case the detection of the go signal is likely to contribute much less variance to the RT, not only because the rise-to-threshold process itself varies much more, but also because the onset of the target serves as a go signal that is highly salient and spatially aligned with the required saccade endpoint.

The results of Peel et al. are nevertheless relevant, in that they highlight another potential source of variance in saccadic RTs (which is now mentioned in the Discussion, subsection “It is all about the base”, third paragraph), albeit one that happens to contribute little under our experimental conditions.